# Perception of microstimulation frequency in human somatosensory cortex

**Christopher L Hughes[1,2,3]\***, **Sharlene N Flesher[1,2,3,4,5]**, **Jeffrey M Weiss[1,6]**, **Michael Boninger[1,2,6,7]**, **Jennifer L Collinger[1,2,3,6,7]**, **Robert A Gaunt[1,2,3,6]\***

[1]Rehab Neural Engineering Labs, University of Pittsburgh, Pittsburgh, United States; [2]Department of Bioengineering, University of Pittsburgh, Pittsburgh, United States; [3]Center for the Neural Basis of Cognition, University of Pittsburgh, Pittsburgh, United States; [4]Department of Neurosurgery, Stanford University, Stanford, United States; [5]Department of Electrical Engineering, Stanford University, Stanford, United States; [6]Department of Physical Medicine and Rehabilitation, University of Pittsburgh, Pittsburgh, United States; [7]Human Engineering Research Laboratories, VA Center of Excellence, Department of Veterans Affairs, Pittsburgh, United States

**Abstract** Microstimulation in the somatosensory cortex can evoke artificial tactile percepts and can be incorporated into bidirectional brain–computer interfaces (BCIs) to restore function after injury or disease. However, little is known about how stimulation parameters themselves affect perception. Here, we stimulated through microelectrode arrays implanted in the somatosensory cortex of two human participants with cervical spinal cord injury and varied the stimulus amplitude, frequency, and train duration. Increasing the amplitude and train duration increased the perceived intensity on all tested electrodes. Surprisingly, we found that increasing the frequency evoked more intense percepts on some electrodes but evoked less-intense percepts on other electrodes. These different frequency–intensity relationships were divided into three groups, which also evoked distinct percept qualities at different stimulus frequencies. Neighboring electrode sites were more likely to belong to the same group. These results support the idea that stimulation frequency directly controls tactile perception and that these different percepts may be related to the organization of somatosensory cortex, which will facilitate principled development of stimulation strategies for bidirectional BCIs.

**\*For correspondence:**
clh180@pitt.edu (CLH);
rag53@pitt.edu (RAG)

**Competing interests:** The authors declare that no competing interests exist.

## Introduction

Bidirectional brain–computer interfaces (BCI) can restore lost function to people living with damage to the brain, spine, and limbs (*Collinger et al., 2018*; *Fetz, 2015*; *Flesher et al., 2021*; *Hughes et al., 2020*). BCI users can control an end effector using neural activity recorded from motor cortex and receive sensory feedback through intracortical microstimulation (ICMS) in somatosensory cortex (*Flesher et al., 2021*). Beyond the practical aim of restoring sensation to improve motor function, existing bidirectional BCIs in human participants provide an unprecedented ability to investigate the nature of sensory perception.

The behavioral effects of ICMS in somatosensory cortex have been studied in detail in non-human primates (NHPs) (*Dadarlat et al., 2015*; *Kim et al., 2015a*; *Kim et al., 2015b*; *Romo et al., 2000*; *Romo et al., 1998*). However, animals are limited in their ability to perform certain psychophysical tasks. NHPs can learn to discriminate between two or more stimuli, and their ability to perform these tasks can provide insight into how stimulus parameters affect sensory perception. However, they can never describe the qualitative nature of the sensory percepts, nor can they be trained to perform more complex psychophysical tasks such as magnitude estimation. NHP studies can therefore lead

to hypotheses about how stimulus parameters affect qualitative aspects of perception, but only human studies can investigate these directly.

Limited work has been conducted in humans using ICMS of somatosensory cortex to restore sensation (*Armenta Salas et al., 2018*; *Fifer et al., 2020*; *Flesher et al., 2016*). From these studies we know that ICMS can evoke tactile sensations that are perceived to originate from the hands (*Fifer et al., 2020*; *Flesher et al., 2016*) and arms (*Armenta Salas et al., 2018*) and that the stimulation locations in the cortex that elicit these percepts agree with known cortical somatotopy (*Penfield and Boldrey, 1937*). Participants reported naturalistic sensations such as 'pressure' and 'touch' (*Flesher et al., 2016*) as well as 'squeeze' and 'tap' (*Armenta Salas et al., 2018*), but the quality and naturalness varied between stimulated electrodes within each participant. Additionally, all studies found that increasing the stimulus current amplitude consistently increased the perceived intensity of the tactile percepts. The effect of stimulus pulse frequency has been less studied, although low frequencies may require higher amplitudes to evoke a detectable percept (*Armenta Salas et al., 2018*).

More is known about the perceptual effects of stimulating the human thalamus (*Davis et al., 1996*; *Dostrovsky et al., 1993*; *Heming et al., 2010*; *Ohara et al., 2004*; *Swan et al., 2018*; *Willsey et al., 2020*). Similar to ICMS, increasing the stimulation amplitude increased percept intensity (*Dostrovsky et al., 1993*; *Swan et al., 2018*). However, changing stimulation frequency and temporal patterns have had different effects on perception. In some cases high-frequency stimulation (333 Hz) evoked the most natural percepts (*Heming et al., 2010*), while in others it evoked primarily paresthesias and low-frequencies produced "tapping" sensations (*Dostrovsky et al., 1993*). In other cases burst stimulation evoked more natural percepts than tonic stimulation (*Willsey et al., 2020*) and two-pulse bursts evoked less natural sensations (*Swan et al., 2018*). Ultimately, temporal factors have clear effects on the sensations evoked through thalamic stimulation, but it remains unclear how to optimally control these parameters to manipulate percept quality.

It has often been suggested that increasing the stimulus frequency increases the perceived intensity of a stimulus train. Increasing the pulse frequency of ICMS reduced the current amplitude required to evoke a detectable percept in NHPs (*Kim et al., 2015a*; *Romo et al., 2000*; *Romo et al., 1998*) and rats (*Butovas and Schwarz, 2007*; *Semprini et al., 2012*). This was thought to indicate that increasing pulse frequency increased perceived intensity. Additionally, in a frequency discrimination task, increasing amplitude biased NHPs (*Callier et al., 2020*) and rats (*Fridman et al., 2010*) toward selecting stimulus trains as having higher frequencies, providing further evidence that increasing pulse frequency increases perceived intensity. Perceived intensity also increases as stimulation amplitude and frequency are increased in human peripheral nerves (*Graczyk et al., 2016*) and human visual cortex (*Schmidt et al., 1996*). This is also true for mechanical stimuli where perceived intensity increased with increasing vibration frequency in able-bodied human participants using tactile input to the hand (*Hollins and Roy, 1996*; *Muniak et al., 2007*; *Verrillo et al., 1969*). Overall, these results imply that both electrical and mechanical stimulation with higher frequency components are perceived as being more intense. Our goal here was to understand whether this same principle applies to ICMS of human somatosensory cortex and to evaluate whether perceptual qualities were affected by changes in stimulus pulse frequency.

In ongoing experiments, we implanted microelectrode arrays into the motor and somatosensory cortices of two participants (referred to as P2 and P3) with cervical spinal cord injuries to evaluate the safety and efficacy of bidirectional BCIs and to study sensorimotor control in humans. In P2, ICMS of somatosensory cortex evoked tactile percepts that felt like they originated from the paralyzed hand (*Flesher et al., 2016*). However, the percepts themselves varied considerably, from more natural sensations, such as touch and pressure, to less natural sensations, such as vibration and tingle. In order to represent more complex and intuitive tactile inputs with ICMS, it is critical that we understand how stimulus parameters directly affect sensation.

We are particularly interested in how stimulus parameters, such as current amplitude, pulse frequency, and train duration, change the perceived intensity of tactile percepts. The ability to control perceived intensity in a bidirectional BCI will be essential, as modulated sensory feedback is crucial for object interaction (*Johansson and Flanagan, 2009*; *Nowak et al., 2013*). While grasp contact could be relayed by simple on–off stimulation, conveying grip force, which is essential for grasp stability, efficiency, and precision (*Godfrey et al., 2016*; *Nowak et al., 2004*; *Nowak and Hermsdörfer, 2006*), requires the ability to modulate the perceived intensity of a stimulus. We sought to

assess the effects of changing the stimulus pulse frequency on several perceptual metrics in two participants, P2 and P3, and expected to see increases in the perceived intensity as the stimulus pulse frequency increased.

## Results

### Effects of frequency on perceived intensity are electrode-dependent

In participant P2, we delivered ICMS trains through individual electrodes and asked him to report the perceived intensity on a self-selected scale, which typically ranged from 0 to 4. We found that increasing the stimulus current amplitude and train duration consistently increased the perceived intensity of the evoked sensations on all tested electrodes (*Figure 1—figure supplement 1*). However, the relationship between stimulus frequency and perceived intensity was electrode dependent (*Figure 1*). We delivered a 60 µA stimulus train for 1 s at pulse frequencies ranging from 20 to 300 Hz. On some electrodes, percept intensity increased with stimulus pulse frequency (*Figure 1B*). However, on over half of the tested electrodes, the opposite effect occurred; stimulus trains with low pulse frequencies (20–100 Hz) were perceived as being the most intense and the intensity *decreased* as the stimulus pulse frequency *increased* (*Figure 1C,D*). We used k-means clustering to separate electrodes into three categories based on the reported percept intensity at 20, 100, and 300 Hz (*Figure 1—figure supplement 2*): electrodes with the highest intensity response at 20 Hz (*Figure 2A*), electrodes with the highest intensity responses at 100 Hz (*Figure 2B*), and electrodes with the highest intensity response at 300 Hz (*Figure 2C*). For simplicity, we refer to these groups based on the pulse frequency range at which the maximal intensity occurred: high-frequency preferring (HFP), intermediate-frequency preferring (IFP), and low-frequency preferring (LFP) electrodes. These electrode groups varied in both the median-reported intensity across all frequencies as well as the frequency at which the maximum intensity occurred.

Seven electrodes were tested multiple times (three to six per electrode) to determine whether the relationships between pulse frequency and perceived intensity were consistent across sessions. The perceived intensity on these electrodes changed by statistically significant amounts as the stimulus pulse frequency changed (p<0.001, Friedman test). The reported intensities at each frequency on these electrodes did not change significantly across test days (p>0.05, Friedman test) (*Figure 1—figure supplement 3*). An additional 22 electrodes were tested in one or two sessions. Of the 29 electrodes tested in total, 20 electrodes exhibited perceived intensities that changed by statistically significant amounts as the stimulus frequency changed (p<0.02, Friedman test). Of these 20 electrodes, five were classified as LFP, seven were classified as IFP, and eight were classified as HFP.

The three different electrode groups had significantly different median intensities (p<0.001, Kruskal–Wallis). Electrodes categorized as IFP had the highest median intensity, while electrodes categorized as HFP had the lowest median intensity (*Figure 1A*).

In participant P3, we tested 23 electrodes at 80 µA and three different frequencies (20, 100, and 300 Hz). The perceived intensity changed by statistically significant amounts on 22 electrodes as the stimulus frequency changed (p<0.05, Friedman test). There were similar electrode-specific effects, where some electrodes evoked the highest intensity percepts at the highest frequencies and others had the highest intensity at the lowest frequencies (*Figure 1—figure supplement 4*). Using the same clustering approach, the data divided into two clusters, which were most similar to the LFP and HFP categories. Fifteen electrodes were classified as HFP, and seven were classified as LFP.

### Frequency-intensity relationships are preserved across suprathreshold amplitudes

We measured whether the frequency–intensity relationships were affected by stimulus current amplitude. If the frequency–intensity relationships were dependent on the current amplitude, this result might reflect idiosyncratic recruitment effects of ICMS. Therefore, in P2, we presented stimulus trains at three current amplitudes (20, 50, and 80 µA) and three pulse frequencies (20, 100, and 300 Hz), which spanned the range of detectable and safe parameters, and asked the participant to report the perceived intensity of the evoked percepts. There were no significant differences in the shape of the frequency–intensity relationships for the three electrode groups at 50 and 80 µA after controlling for changes in median intensity caused by increasing current amplitude (p=0.21–0.99, Friedman's test,

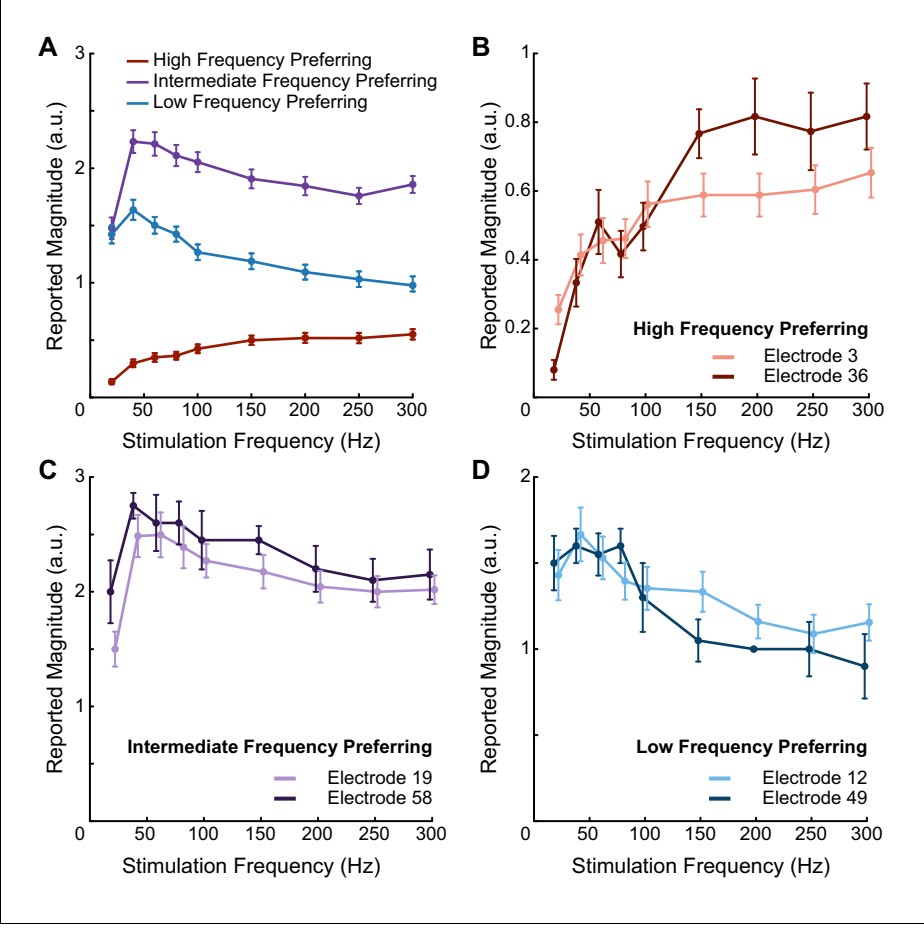

**Figure 1.** Pulse frequency drives electrode-specific changes in intensity which can be grouped into three categories. (A) Perceived intensity for each aggregated frequency preference group. Different colors represent different categories. Each data point shows the mean intensity response of all of the electrodes in a given category. (B) Perceived intensity for two examples of high-frequency preferring electrodes that evoked the most intense percepts at the highest pulse frequencies and that generated the least intense percepts overall. (C) Perceived intensity for two examples of intermediate-frequency preferring electrodes that generated the most intense overall percepts, which occurred between 40 Hz and 100 Hz. (D) Perceived intensity for two examples of low-frequency preferring electrodes, which generated intermediate overall intensities that were maximized between 20 and 100 Hz. Error bars represent the standard error. The points are connected with piecewise fits. Axes are scaled differently between panels for clarity.

The online version of this article includes the following source data and figure supplement(s) for figure 1:

**Source data 1.** This file contains all the magnitude estimation data from participant P2 using an amplitude of 60 µA and frequencies of 20, 40, 60, 80, 100, 150, 200, 250, and 300 Hz.

**Figure supplement 1.** Increases in current amplitude and train duration consistently drive increases in perceived intensity.

**Figure supplement 1—source data 1.** This file contains the data from participant P2 for magnitude estimation using a frequency of 100 Hz.

**Figure supplement 2.** Electrodes divide into three categories based on their frequency–intensity relationships.

**Figure supplement 2—source data 1.** This file contains the mean reported intensity and standard error for participant P2 for magnitude estimation trials using an amplitude of 60 µA and frequencies of 20, 40, 60, 80, 100, 150, 200, 250, and 300 Hz.

**Figure supplement 3.** Electrodes maintain same frequency–intensity relationships over time.

**Figure supplement 3—source data 1.** This file contains the normalized median-reported intensity for participant P2 for magnitude estimation trials using an amplitude of 60 µA and frequencies of 20, 40, 60, 80, 100, 150, 200, 250, and 300 Hz.

**Figure supplement 4.** Electrode-specific frequency–intensity relationships and spatial clustering generalize to a second participant.

*Figure 1 continued on next page*

*Figure 1 continued*

**Figure supplement 4—source data 1.** This file contains all the magnitude estimation data from participant P3 using an amplitude of 80 µA and frequencies of 20, 100, and 300 Hz.

*Figure 2*). The reported intensity on LFP electrodes peaked at 20 Hz at both current amplitudes (p=0.02, Kruskal–Wallis, *Figure 2A*), whereas the reported intensities of IFP electrodes peaked at 100 Hz for both current amplitudes (p<0.001, Kruskal–Wallis, *Figure 2B*) and the reported intensity on HFP electrodes peaked at 300 Hz for both current amplitudes (p<0.001, Kruskal–Wallis, *Figure 2C*). Interestingly, when we decreased the current amplitude to 20 µA, which was close to the detection threshold for most electrodes, increasing the pulse frequency from 20 to 100 Hz evoked more intense percepts for all electrode groups (p<0.05, Kruskal–Wallis, *Figures 2A–C*, 20 µA). There were highly significant differences between the shape of the frequency–intensity relationships for all groups at 20 µA versus 50 or 80 µA (p<0.001, Friedman's test) even after controlling for changes in the median intensity caused by increasing current amplitude. At 20 µA, the percept intensity was very low, making magnitude estimation akin to a detection task.

## High-frequency stimuli are detected more reliably at perithreshold amplitudes

Our observation that higher stimulus pulse frequencies can evoke less-intense percepts at suprathreshold stimulus current amplitudes differs from predictions made from non-human primate studies; higher frequencies evoked detectable percepts at lower amplitudes in NHPs, which led to predictions that higher frequency always results in higher perceived intensities (*Kim et al., 2015a*; *Romo et al., 2000*; *Romo et al., 1998*). However, the effect of changing ICMS parameters on perceived intensity cannot be tested directly in NHPs. Indeed, we found that the perceived intensity at

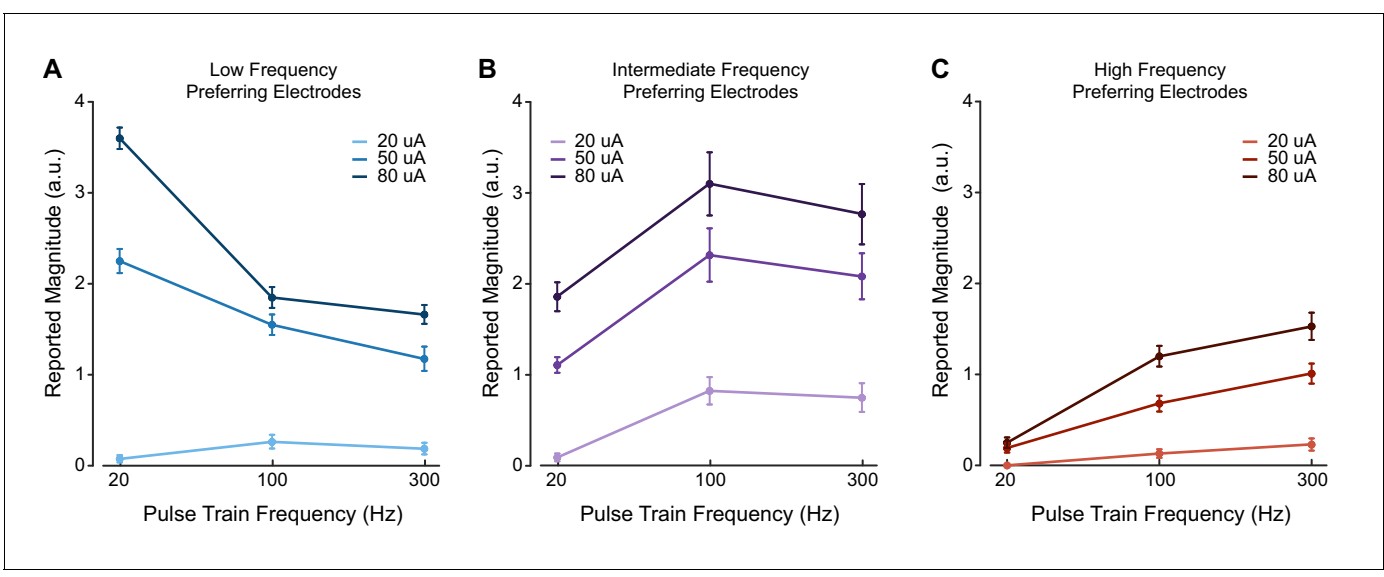

**Figure 2.** Stimulus current amplitude does not change the relationship between pulse frequency and intensity at suprathreshold amplitudes. Magnitude estimation data for different current amplitudes and pulse frequencies. Data were aggregated across electrodes by their category, where each plot shows a different category of electrodes. Perceived intensity values for (**A**) LFP electrodes, (**B**) IFP electrodes, and (**C**) HFP electrodes at different current amplitudes and pulse frequencies. Different colored bars represent different current amplitudes. Error bars indicate the standard error across electrodes. We tested two LFP electrodes, three IFP electrodes, and two HFP electrodes which were each tested twice in different sessions. The online version of this article includes the following source data and figure supplement(s) for figure 2:

**Source data 1.** This file contains all the magnitude estimation data from participant P2 using amplitudes of 20, 50, and 80 µA and frequencies of 20, 100, and 300 Hz.

**Figure supplement 1.** Higher pulse frequencies always improved detection at perithreshold current amplitudes.

**Figure supplement 1—source data 1.** This file contains all the data from participant P2 for a detection task conducted at perithreshold amplitudes.

the lowest tested currents always increased when the frequency increased from 20 to 100 Hz (*Figures 2A–C*, 20 µA), but that this effect was not always maintained at higher current amplitudes (*Figures 2A,B*, 50 and 80 µA). To explicitly compare our results to NHP work, we performed a detection task in P2 in which the current amplitude was set to perithreshold levels and the pulse frequency was varied between 20, 100, and 300 Hz. We found that at 300 Hz, the interval containing the stimulus train was correctly identified 80% of the time across all tested electrodes (*Figure 2—figure supplement 1*). Similarly, when the pulse frequency was set to 100 Hz, the mean detection accuracy was 72%. In contrast, when the pulse frequency was set to 20 Hz, the mean detection accuracy was just 42%, which was not significantly different than chance levels of 50% (p=0.14, one-sample t-test). Detection accuracies at 100 Hz and 300 Hz were significantly higher than the detection accuracy at 20 Hz (p<0.05, ANOVA) but were not significantly different from each other (p=0.66, ANOVA).

## Frequency-intensity relationships are associated with different perceptual qualities

One advantage of studying somatosensation in humans is the ability to document the sensory qualities evoked by stimulation (*Figure 3—figure supplement 1*). We found that there were significant differences in the qualities evoked on electrodes belonging to different categories defined by the effect of pulse frequency on intensity in P2 (*Figure 3A*). Additionally, the sensory qualities for electrodes in each group were differentially modulated by pulse frequency (*Figure 3B*).

At 20 Hz, LFP and IFP electrodes evoked percepts with pressure, tapping, sparkle, and touch qualities. These qualities were not evoked on HFP electrodes at any frequency. At this low stimulation frequency, HFP electrodes were generally not detectable, resulting in few reports of any percepts. At 100 Hz, IFP electrodes evoked percepts with buzzing, vibration, and sharp qualities. LFP and HFP electrodes never evoked these qualities when stimulated at 100 Hz. HFP electrodes also evoked sensations of touch and prick at 100 Hz that never occurred on LFP or IFP electrodes at any frequency. However, these qualities occurred on less than 30% of trials on HFP electrodes. At 300 Hz, the responses were similar to those at 100 Hz except that all electrode categories evoked less pressure.

We also clustered electrodes based on the verbal reports of percept quality at all frequencies. Interestingly, these clusters were remarkably similar to those based on intensity responses at different frequencies (*Figure 3—figure supplement 2*). That these electrode categories were nearly identical when created using completely different data sets – perceptual qualities and perceived intensities – strongly suggests that these two features are measures of the same underlying properties of the neurons recruited by stimulation.

## Perceptual responses are spatially clustered in cortex

Finally, we asked whether the categorization of an electrode, which corresponds to its frequency–intensity responses and evoked perceptual qualities, was related to its location in cortex. We compared the observed spatial occurrence of the different electrode categories with a simulation that randomly assigned each category to one of the tested electrode locations while maintaining the same number of electrodes in each category. In P2, there was significant clustering of electrodes in the same category (*Figure 4A*) across arrays (pseudo-p=0.00017). This was particularly apparent on the lateral array. In P3, LFP electrodes only occurred on one of the arrays (*Figure 4—figure supplement 1*), which resulted in clustering across the arrays (pseudo-p=0.0045, local indicators of spatial association [LISA]).

While there was some overlap between the projected field location and frequency preference, in some cases, electrodes with different frequency preferences evoked percepts from the same region of the hand (*Figure 4B*). For example, LFP, IFP, and HFP electrodes elicited sensations on the palmar region beneath the middle and ring fingers. As a result, percepts from a single region of the hand could be evoked by electrodes that generated multiple response types.

## Discussion

We found that ICMS frequency alters the perceived intensity (*Figure 1*, *Figure 1—figure supplement 4*) and quality (*Figure 3*) in an electrode-specific manner. Furthermore, we found that electrodes with similar intensity responses and qualities clustered spatially in somatosensory cortex

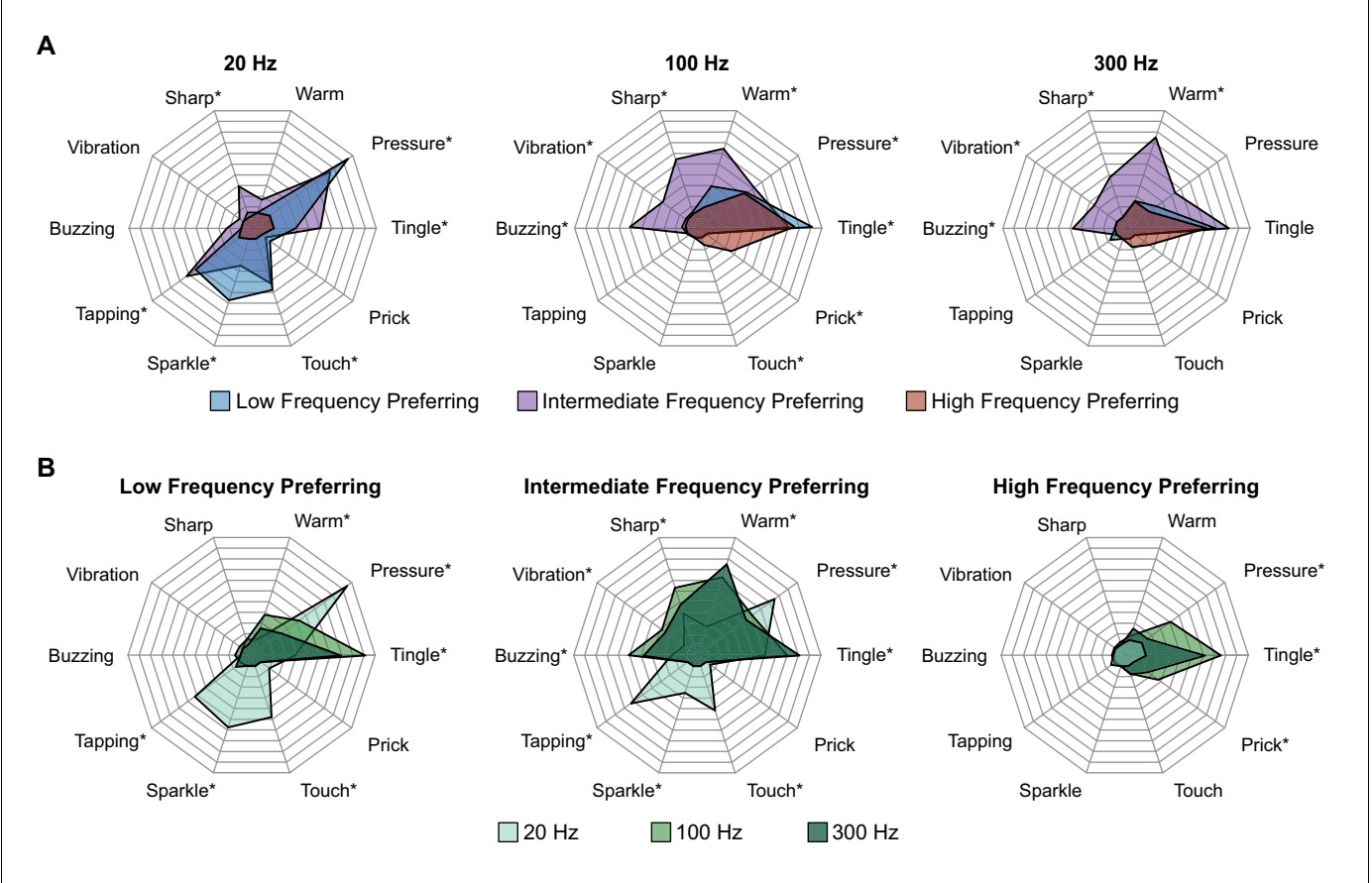

**Figure 3.** Perceptual qualities are associated with specific electrode categories and stimulus pulse frequencies. Radar plots showing the distribution of reported qualities at different pulse frequencies for each electrode category. (**A**) Percepts sorted by pulse frequency. Electrode categories are indicated with different colors. (**B**) Percepts sorted by electrode categories. Pulse frequencies are indicated with different colors. In each plot, qualities on which there was a significant difference between categories, as determined with Fisher's exact test, are marked with an asterisk.

The online version of this article includes the following source data and figure supplement(s) for figure 3:

**Source data 1.** This file contains the total number of reports of each percept quality in participant P2 across each frequency preference group (LFP, IFP, and HFP).

**Figure supplement 1.** All reported percepts and their percent occurrence at each pulse frequency.

**Figure supplement 1—source data 1.** This file contains the percept identifiers from the perceptual reports from the surveys from P2.

**Figure supplement 2.** Clustering by evoked qualities results in nearly identical clusters to those identified from perceived intensity.

**Figure supplement 2—source data 1.** This file contains the median intensities at 20, 100, and 300 Hz reported by participant P2 for each electrode tested as well as the cluster number that was assigned by k-means clustering based on the qualitative data.

(*Figure 4*, *Figure 4—figure supplement 1*). This implies that the observed electrode-specific relationships between frequency and perception are not caused by random factors and are instead related to the underlying structure of the cortex.

## Neural populations preferentially respond to different stimulus frequencies in somatosensory cortex

Intracortical microstimulation at the maximum amplitudes used in this study can directly activate neurons up to 2 mm away from the electrode tip, but most activation occurs less than 500 µm from the electrode tip (*Overstreet et al., 2013*; *Stoney et al., 1968*). At intermediate amplitudes (e.g. 50–60 µA), direct activation primarily occurs within 200–300 µm of the electrode tip. Stimulation can also recruit passing axons which can project to far away areas, resulting in sparse, distributed activation of the cortex (*Histed et al., 2009*; *Michelson et al., 2019*). Using optical imaging, clusters of neurons with similar responses extend from 0.2 to 1 mm in squirrel monkeys (*Friedman et al., 2004*)

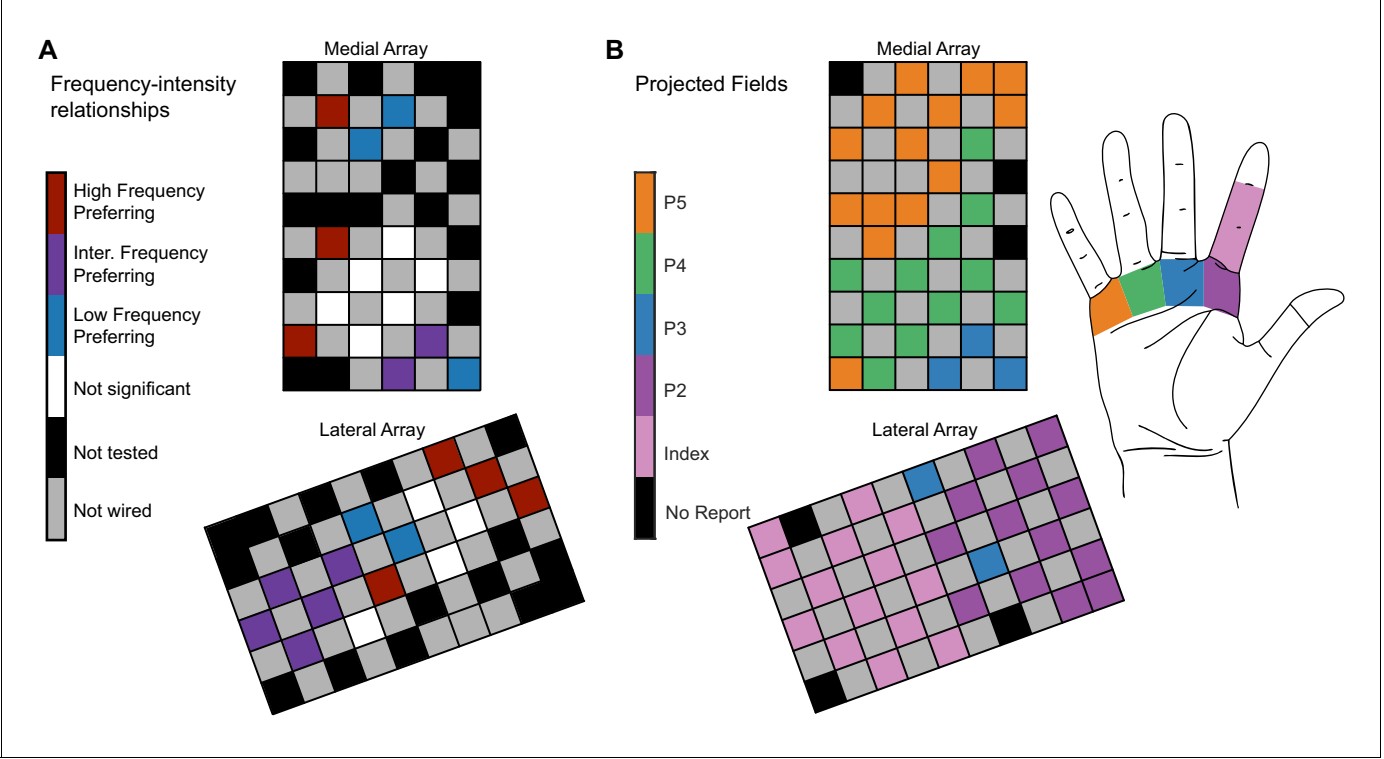

**Figure 4.** Electrode location is significantly related to electrode categorization. (**A**) Map of the medial electrode array (top) and lateral electrode array (bottom) implanted in somatosensory cortex and the distribution of the frequency preference categorizations. The electrode arrays were implanted close to the central sulcus with the left edge of the medial array being approximately parallel to the central sulcus. The arrays are oriented to reflect the implant orientation. Colored squares represent different types of electrodes as indicated by the color bar. (**B**) The projected field locations for each tested electrode. The label for each electrode corresponds to the most reported projected field for each electrode on all 100 Hz surveys taken in the same year as the magnitude estimation data.

The online version of this article includes the following source data and figure supplement(s) for figure 4:

**Source data 1.** This file contains the spatial mapping of each electrode and the frequency preference group for each electrode for participant P2.
**Figure supplement 1.** The spatial mapping of the two groups on the arrays for P3.
**Figure supplement 1—source data 1.** This file contains the spatial mapping of each electrode and the frequency preference group for each electrode in participant P3.

while electrophysiological recordings have measured similar effects over 0.5 to 1 mm in the medio-lateral direction and multiple millimeters in the rostrocaudal direction in macaque monkeys (*Sur et al., 1984*). These spatial scales over which function varies are similar to the expected recruitment distances from ICMS, supporting the idea that different perceptual or frequency responses may be linked to activating different functional groups of neurons.

Electrophysiological (*Mountcastle et al., 1969*; *Sur et al., 1981*; *Sur et al., 1984*) and optical (*Chen et al., 2001*; *Friedman et al., 2004*) recordings have shown organized neural populations in the somatosensory cortex that are sensitive to tactile input with specific frequency content. These experiments promoted the idea of submodality separation in the cortex in which the activity of cortical neurons is primarily driven by input from either rapidly adapting Meissner corpuscles (RAs), slowly adapting Merkel cells (SAs), or Pacinian corpuscles (PCs). However, many cortical neurons receive heterogeneous input from multiple classes of mechanoreceptors (*Pei et al., 2009*; *Reed et al., 2010*; *Saal and Bensmaia, 2014*), resulting in neurons that can exhibit both sustained and transient responses. Therefore, the different effects of stimulus frequency on intensity and perception are unlikely to arise from activating inputs representing specific tactile subpopulations (e.g. SAs, RAs, or PCs), but instead from how a local region of somatosensory cortex can respond to different stimulus frequencies, consistent with the concept of the cortex encoding different frequency features (*Prsa et al., 2019*).

The idea that somatosensory cortex is organized for feature encoding is supported by human psychophysics experiments where frequency perception was dependent on specific spiking patterns and not on the types of mechanoreceptor that were activated (*Birznieks et al., 2019*). Similarly, individual cells in mouse cortex are preferentially activated by different mechanical stimulation frequencies (*Prsa et al., 2019*). In those same experiments, the frequency preference of the neural population tended toward higher frequencies when the indentation depth decreased, similar to our results that higher frequencies were perceived as being more intense when the ICMS amplitude was decreased (*Figure 2*). Together, these results suggest that the somatosensory cortex receives convergent input from different mechanoreceptors and is organized for feature-selective encoding, which results in different preferential responses to ICMS frequency and different evoked qualities.

## Mechanisms for heterogeneous perceptual responses to stimulus frequencies in cortex

The effects described here must be related to different cellular responses to stimulation in different regions of the cortex. In fact, different stimulation frequencies in mouse somatosensory cortex can alter the activation of neurons far away from the stimulation electrode (*Michelson et al., 2019*). Specifically, high pulse frequencies lead to rapid habituation of neurons far away from the electrode, while low pulse frequencies can maintain the activation of these same neurons. This reduced activity in neurons far away from the electrode could lead to decreases in perceived intensity and changes in percept quality in a way that depends on electrode location and local neural populations.

A potential mechanism to explain electrode-dependent responses are varying distributions of inhibitory and excitatory neurons. The presence of more inhibitory neurons in a local region could result in stronger inhibitory drive at higher frequencies, resulting in more robust responses to low-frequency stimuli. Indeed, recruitment of inhibitory Martinotti cells in the somatosensory cortex of rats increases as the duration and frequency of presynaptic action potentials increase (*Kapfer et al., 2007*; *Silberberg and Markram, 2007*). Furthermore, rostrocaudal heterogeneity of inhibition has been documented in rat olfactory cortex (*Large et al., 2018*; *Luna and Pettit, 2010*). Whether such organization exists in human somatosensory cortex remains to be seen.

Short-term plasticity (*Tsodyks and Markram, 1997*) at synapses driven by stimulation may also explain the observed effects. If a synapse is unable to resupply neurotransmitter at a rate faster than the stimulus frequency, transmitter release at the synapse could become depressed. In this scenario, neurons would be unable to recruit other neurons in synchrony with stimulation, which could result in lower recruitment and lower perceived intensity. If cells in somatosensory cortex have different time constants for transmitter recovery, this could serve as a mechanism for frequency filtering (*Rosenbaum et al., 2012*). Elucidating the precise mechanisms underlying observed frequency responses in cortex will require further studies in animal models.

## ICMS in humans directly evaluates intensity and perception

Higher stimulus pulse frequencies decreased the current amplitude required to detect a stimulus train in NHPs (*Kim et al., 2015a*). This suggested that higher stimulus frequencies would increase the perceived intensity of a stimulus train. Similar to these animal studies, we found that higher frequencies improved the detectability of stimulus trains at perithreshold amplitudes. However, at suprathreshold current amplitudes, increasing the frequency did not always produce higher perceived intensities. A question that emerges then is whether the prediction of increasing intensities at higher frequencies can be reconciled with our observations of decreased intensities at higher frequencies on a subset of the electrodes.

To determine whether changes in frequency could be perceived independently of changes in amplitude, animals were trained to identify which of two intervals contained the higher frequency stimulus train, regardless of current amplitude (*Callier et al., 2020*). Increasing the amplitude always biased the animals toward selecting a stimulus train as having a higher frequency, suggesting that both amplitude and frequency have similar perceptual effects. However, animals were still able to distinguish between changes in amplitude and frequency on some electrodes. In our experiments, LFP and IFP electrodes, which generated high-intensity percepts at low frequencies, often evoked percepts with highly salient qualities, such as tapping or buzzing. The presence of these qualities at some frequencies and not others (*Figure 3*) would allow the participant to distinguish between

increases in amplitude, which only increases the percept intensity (*Figure 1—figure supplement 1*), and increases in frequency, which changes the percept quality and intensity (*Figure 3*). On electrodes without highly salient frequency-dependent qualities, such as the HFP electrodes, it would be difficult to disambiguate changes in amplitude and frequency.

However, an important difference between these experiments is that many electrodes in our study evoked less-intense percepts as the pulse frequency increased, which was not observed in NHPs. The reason for this is unclear, and it may be related to the larger frequency range explored in this study or the electrode location in the cortex. Another interpretation is that since frequency can change percept quality (*Figure 3*), different qualities are understood to have different intensities. Animals cannot directly report perceived intensity on an open scale as is simply done in humans. Rather, perceived intensity, as well as other subjective aspects of perception such as quality and naturalness, must be inferred from other experimental paradigms, which makes it difficult to assess how ICMS affects subjective aspects of perception in animals. This demonstrates that human experiments are crucial to understanding how ICMS modulates tactile perception, particularly for subjective evaluation of experience.

## Limitations of study

There are several limitations associated with this work. First, most of these experiments were conducted in a single participant with a chronic implant. Different participants, with different timelines of injury preceding implant, could potentially respond differently to stimulation, particularly if the electrodes are implanted in a different part of the somatosensory cortex. However, the repeatability of our findings suggests that these effects were at least not due to day-to-day variations. Additionally, we found electrode-specific frequency effects, including LFP electrodes, that were spatially clustered in a second participant. This suggests that changing frequency will affect intensity and perception similarly in other participants. One important difference in the second participant was that we only observed LFP relationships on a single array.

The participants had limited residual sensation in their hands, which made it difficult to measure responses in cortex to tactile indentation. Comparing perceptual responses to ICMS with cortical responses to tactile indentation could help better relate these findings to previous studies in monkeys. Additionally, it is notable that due to spinal cord injury there may be reorganization of cortex (*Chen et al., 2002*; *Freund et al., 2011*; *Henderson et al., 2011*; *Wrigley et al., 2009*). However, recent work has argued that measured remapping may be simply driven by the uncovering of pre-existing latent activity, corresponding instead to homeostasis (*Makin and Bensmaia, 2017*; *Muret and Makin, 2021*). The ability to elicit sensations with ICMS years after injury is supportive of this idea (*Armenta Salas et al., 2018*; *Fifer et al., 2020*; *Flesher et al., 2016*).

Another potential confound is that perceived intensity can change throughout a session. Because we used pseudo-randomized presentations of different stimulus parameters to ensure that electrodes were not tested in the same order, and excluded the first block of trials from analysis for each set for magnitude estimation, we believe that this phenomenon did not affect our results.

Our results are consistent with the idea that somatosensory cortex is organized in a way that represents different features in different locations; however, there are several limitations that should be considered. First, the electrodes covered just a small region of somatosensory cortex, and with a limited spatial resolution, limiting the ability to create detailed maps. Second, we divided electrodes into three groups for participant P2 and two groups for participant P3. This categorical division arose from considering the frequency–intensity relationships and the unique perceptual qualities reported for the electrodes in each group. Categorical divisions are commonly used to describe neural responses in the cortex, including somatosensory cortex (*Friedman et al., 2004*; *Sur et al., 1981*; *Sur et al., 1984*). However, neurons receive convergent input from multiple sub-type modalities (*DiCarlo et al., 1998*; *Saal and Bensmaia, 2014*), and it is possible that the responses to stimulation may divide into more than three groups or fall on a spectrum of different frequency preferences with no discrete categories. More data will need to be collected across additional participants and regions of somatosensory cortex to see whether these patterns persist. Third, we do not know if electrodes across the array are in different layers of cortex. Different layers of cortex may drive different perceptual responses with the same input. However, if this were the case, this would still reflect important functional differences in cortex, which need to be understood for bidirectional BCIs.

Finally, a challenge for developing mechanistic explanations of these observations is that there are few neuroscientific tools that we can use to further probe these effects in a human. Because of this, addressing the neurophysiological mechanisms of these frequency responses is difficult in a human participant, and further investigation of these properties in animal models is needed.

### Implications for prostheses

Stimulus amplitude linearly modulates intensity, while frequency has non-monotonic and electrode-specific effects on intensity and percept quality. To signal changing the intensity of a tactile input, amplitude should be used and not frequency. Other potential options also exist to modulate intensity that were not explored in this paper, including pulse width modulation and multielectrode stimulation. Future studies should assess the efficacy of these parameters.

Knowing that different electrodes encode different perceptual features can inform our approach to creating a functional bidirectional BCI in two primary ways. First, these results may help identify electrodes that have perceptual or intensive properties that are relevant to the task being performed. Certain electrodes are more likely to represent specific perceptual qualities, and these electrodes could be selectively used depending on the type of tactile input to the prosthetic device.

Second, these results suggest that electrode-specific stimulation encoding schemes would be particularly useful. In the peripheral nervous system, biomimetic approaches to stimulation using models such as TouchMime have been used (*George et al., 2019*; *Okorokova et al., 2018*; *Valle et al., 2018*). In the cortex, combining these biomimetic models with electrode selection based on measured feature-preferences may yield more natural percepts. For example, electrodes that represent 'tapping' sensations could receive large amplitude transients, signaling the onset and offset transients, while electrodes that do not evoke this sensation could receive low-amplitude, tonic stimulation, signaling maintained contact. Another promising future direction is to use machine learning methods to categorize the feature-preferences of different electrodes more quickly and accurately. These methods could ultimately improve the usefulness of somatosensory feedback, in turn improving the performance of bidirectional BCIs and ultimately improving the quality of life for people living with paralysis.

## Materials and methods

### Regulatory and subject details

This study was conducted under an Investigational Device Exemption from the U.S. Food and Drug administration, approved by the Institutional Review Boards at the University of Pittsburgh (Pittsburgh, PA) and the Space and Naval Warfare Systems Center Pacific (San Diego, CA), and registered at ClinicalTrials.gov (NCT0189-4802). Informed consent was obtained before any study procedures were conducted. The purpose of this trial is to collect preliminary safety information and demonstrate that intracortical electrode arrays can be used by people with tetraplegia to both control external devices and generate tactile percepts from the paralyzed limbs; this manuscript presents the analysis of data that were collected during participation in the trial but does not report clinical trial outcomes. All data included in this paper (including magnitude estimation, surveys, detection thresholds, etc.) were limited to 1 year of data collection in P2 to minimize the impact of changes in perception that can occur over long time periods. Data in P3 were collected over 2 months.

Participant P2 was 28 years old at the time of implant and had a C5 motor/C6 sensory ASIA B spinal cord injury. Two microelectrode arrays (Blackrock Microsystems, Salt Lake City, UT) were implanted into the somatosensory cortex. Results from this participant have been reported previously (*Flesher et al., 2016*; *Flesher et al., 2021*; *Hughes et al., 2021a*). Each electrode array consisted of 32 wired electrodes arranged on a 6 × 10 grid with a 400 μm interelectrode spacing resulting in a device with an overall footprint of 2.4 × 4 mm. The remaining 28 electrodes were not wired due to technical constraints related to the total available number of electrical contacts on the percutaneous connector. Electrode tips were coated with a sputtered iridium oxide film. The stimulation return electrode was the titanium pedestal that was fixed to the skull.

Participant P3 was 28 years old at the time of implant and had a C6 ASIA B spinal cord injury. He received the same type of microelectrode arrays in the somatosensory cortex. Data from this participant have not been published previously. The electrodes were also targeted to the hand region of

area 1 of the somatosensory cortex using preoperative imaging and evoked sensations that he described as originating from his hand.

## Stimulation protocol

Stimulation was delivered using a CereStim C96 multichannel microstimulation system (Blackrock Microsystems, Salt Lake City, UT). Pulse trains consisted of cathodal phase first, current-controlled, charge-balanced pulses, which could be delivered at frequencies from 20 to 300 Hz and at amplitudes from 2 to 100 µA. The cathodal phase was 200 µs long, the anodal phase was 400 µs long, and the anodal phase was set to half the amplitude of the cathodal phase. The phases were separated by a 100 µs interphase period. At the beginning of each test session involving stimulation, we sequentially stimulated each electrode first at 10 µA and 100 Hz for 0.5 s and then at 20 µA and 100 Hz for 0.5 s. During these trials, the interphase voltage on each electrode was measured at the end of the interphase period, immediately before the anodal phase (*Cogan, 2008*). If an electrode's measured interphase voltage was less than −1.5 V, the electrode was disabled for the day (*Flesher et al., 2016*). This step was performed to minimize stimulation on electrodes that might potentially experience high voltages, which could result in irreversible damage.

## Magnitude estimation

We assessed the effect of stimulus parameters on perceived intensity using a magnitude estimation task. To test the potential effect of pulse frequency on intensity in P2, pulse trains were delivered for 1 s at 60 µA with frequencies of 20, 40, 60, 80, 100, 150, 200, 250, and 300 Hz. Following each pulse train, P2 was asked to report the magnitude of the perceived intensity on a self-selected scale. P2 was instructed to use values such that a value twice as large as a previous value was twice as intense, and a value half as large was half as intense. These values typically ranged from zero to six. Each set of stimulus pulse frequencies was presented six times, with the presentation order randomized in each block. The responses from the first block were not used in the analysis to allow the participant to establish a baseline for reporting for the session. Data collected on the same electrode over multiple sessions were aggregated for analysis. We tested 29 total electrodes using this paradigm. Seven electrodes were tested in three to six sessions, while 22 electrodes were tested in one to two sessions.

To increase the number of trials and decrease the time for data collection, we presented 20, 100, and 300 Hz stimulus trains at 80 µA to participant P3. We presented each frequency 21 times and removed the first trial from the analysis. Twenty-two of the 23 tested electrodes showed a significant difference between intensities across tested frequencies (Friedman's test, p<0.05). Data for each electrode were only collected once.

We also assessed the effect of changing the stimulus current amplitude on perceived intensity, while the stimulus pulse frequency was held constant in P2. The pulse frequency was set to 100 Hz, the train duration to 1 s, and the current amplitude ranged from 20 to 80 µA in 10 µA increments. Data were fit with a linear function. We tested nine electrodes for this paradigm. Finally, we assessed the effect of changing the stimulus train duration on perceived intensity in P2. The stimulus pulse frequency and current amplitude were set to 100 Hz and 60 µA, respectively, and the train duration was set to 0.1, 0.2, 0.3, 0.4, 0.5, 0.75, 1, 1.5, and 2 s. Data were fit with a logistic function. We tested four electrodes for this paradigm. For current amplitude and train duration plots, the data were normalized to the median intensities of the set in which it was collected for visualization purposes.

To investigate the interaction between current amplitude and pulse frequency, we additionally tested frequency and amplitude pairs in P2. The train duration was set to 1 s, the current amplitude was set to 20, 50, or 80 µA, and the pulse frequency was set to 20, 100, or 300 Hz. All frequency and amplitude combinations were tested for each tested electrode six times, and the first trial was excluded from analysis. Each tested electrode was tested twice on two different test sessions, resulting in 10 total trials for each frequency and amplitude pair. For analysis and plotting, we divided electrodes into the categories defined in the frequency magnitude estimation described previously. We tested two LFP electrodes, three IFP electrodes, and two HFP electrodes. We tested six electrodes for this paradigm, each measured twice.

## Detection thresholds

Detection thresholds were determined using a two-alternative forced choice task in P2. P2 was instructed to focus on a fixation cross on a screen located in front of him. Two 1-s-long windows, separated by a variable delay period, which averaged 1 s in length, were presented and indicated by a change in the color of the fixation cross. Stimulation was randomly assigned to one of the two windows. After the last window, the fixation cross disappeared, and the participant was asked to report which window contained the stimulus.

A one-up three-down staircase method was used, so that if the participant correctly identified the window containing the stimulus in three consecutive trials, the current amplitude was decreased for the next trial (*Leek, 2001*; *Levitt, 1971*). If the participant incorrectly identified the window containing the stimulus, the current amplitude was increased for the next trial. The current amplitude started at 10 µA and was increased or decreased by a factor of 2 dB. The pulse frequency was held constant at 100 Hz. This method reduced the time spent testing uninformative values but does not guarantee that all current amplitudes will be tested the same number of times. After five changes in the direction of the stimulus current amplitude (increasing to decreasing, or decreasing to increasing), the trial was stopped. The detection threshold was calculated as the average of the last 10 values tested before the fifth direction change.

We also conducted standard detection trials where the stimulus pulse frequency was changed while the stimulus current amplitude was held constant in P2. The current amplitude was set to 1.2× the detection threshold for each electrode measured at 100 Hz. The tested frequencies were 20, 100, and 300 Hz, and each pulse frequency was presented 30 times. Pulse frequencies were interleaved randomly resulting in 90 trials per tested electrode. We tested four electrodes with this paradigm.

## Surveys

Surveys were conducted once every month from the time the arrays were implanted in P2. During a survey, each enabled electrode was stimulated sequentially using a 1 s pulse train at 60 µA. These parameters were selected because they were typically able to evoke sensations consistently while remaining well below our maximum stimulus current amplitude of 100 µA. In participant P2, surveys were conducted once a month at 100 Hz, but we collected additional surveys at 20 and 300 Hz. This resulted in 152 samples at 20 Hz, 621 samples at 100 Hz, and 85 samples at 300 Hz. Surveys were conducted to quantify stimulus-evoked tactile percepts. No visual or auditory cue was provided to the participant to indicate when stimulation was occurring. The participant was instructed to indicate when a sensation was detected, at which point progression through the trial was paused. The participant verbally reported when he detected a sensation, and the pulse train was repeated as many times as necessary for the participant to be able to accurately describe the location and quality of the sensation. A drawing of the hand was partitioned into different segments and the participant reported on which segments the sensation was felt. The participant also used a tablet and stylus to circumscribe the precise areas where sensation was felt on a map of the hand.

After the location of the percept was established, the participant reported the quality of the sensation using the descriptors in *Figure 3—figure supplement 1*. The participant's response was documented by the experimenter, and video recordings were also taken during all responses. If the participant felt that the sensation was not accurately described by the provided descriptors, his response was recorded, and the best approximation using the descriptors was used. The descriptors included a five-point scale for naturalness ranging from totally unnatural to totally natural, the location of the sensation on or below the skin surface, and an assessment of pain ranging from 0 to 10. The quality of the sensation was further assessed using the following descriptors: mechanical (touch, pressure, or sharp), movement (vibration or movement across the skin), temperature (warm or cool), and tingle (electrical, tickle, or itch). These descriptors were based on a previously described questionnaire (*Heming et al., 2010*). The participant could report multiple qualities for a single stimulus, and in some cases, the subcategories (e.g. electrical, tickle, or itch) could not be described. P2 also reported qualities that deviated from the descriptors. P2 developed four new descriptors that were not originally included, which often were combinations of the other descriptors. We attempted to reidentify these percepts in the context of a new questionnaire, which was published during this study in consultation with the participant (*Kim et al., 2018*). Three of these sensations were

reidentified as 'tapping', 'buzzing', and 'prick'. One descriptor P2 reported, 'sparkle,' could not be reidentified with the new questionnaire. P2 described this percept as feeling like tapping that varied in intensity and moved around the projected field in a random manner. It should be noted that all percepts in our study were identified as tactile percepts and no proprioceptive sensations were evoked.

The survey data collected in P2 included in these analyses were collected during the same year as the frequency magnitude estimation data to ensure the evoked sensations were consistent across paradigms, which included data from post-implant days 630–962.

## K-means clustering

Electrodes were divided into three categories using k-means clustering using the reported intensity at 20, 100, and 300 Hz. Both silhouette and elbow analysis were used to validate that k = 3 was a suitable parameter choice for P2. We labeled the categories as LFP, IFP, and HFP based on the frequency at which the maximum intensity occurred. Based on silhouette analysis, we found that data from P3 divided best into two clusters. We labeled these clusters as LFP and HFP in line with the classification from the first participant.

Electrodes were additionally clustered based on the reported perceptual qualities at 20, 100, and 300 Hz in P2. Each reported quality (of which there were 10) was summed across sessions and pulse frequencies for each electrode. The total number of reports for each quality was then divided by the maximum number of reports for any electrode, so that each quality was represented by number between zero and one and contributed equally to the clustering of each electrode. No dimensionality reduction was used and electrodes were clustered within the 10 dimensions of reported qualities.

## Statistics

All quantification and statistical analyses were done in MATLAB (Mathworks, Natick, MA). Sample sizes are listed in the methods for each experiment. A power analysis was not conducted to determine the number of replicates for each experiment. The number of repetitions for psychophysics experiments were based on commonly used values. Electrodes that elicited clearly perceptible sensations and showed a significant change in perception with a change in a parameter were collected across multiple sessions to determine whether effects were consistent over time.

For all statistical tests, we determined whether to use parametric or non-parametric tests based on the normality of the data as assessed with an Anderson–Darling test. If the data were significantly different than normal, then we used non-parametric tests. Any time multiple comparisons were made, we used the Benjamini–Hochberg procedure to correct for multiple comparisons, which resulted in a critical p-value that was used as a cut-off. If no values were significant, then the critical p-value returned is 0 and not reported and no values are considered significant. For any tests that required post-hoc comparisons, we used Tukey's HSD test.

For magnitude estimation data, we used Friedman's test to assess significant differences between the intensity responses at different pulse frequencies as well as differences between electrode responses across days. Friedman's test also allowed us to compare significant effects of pulse frequency on intensity across multiple sessions by excluding experimental day as a cofactor. When comparing the same electrode across sessions, we compared intensity responses with the same tested pulse frequency and corrected for multiple comparisons. We compared differences in the median intensity of electrodes within each category using a Kruskal–Wallis test.

For detection data, we used an ANOVA to assess significant differences in the detection accuracy at different pulse frequencies.

For quality data obtained from surveys, we used Fisher's exact test to evaluate whether there was a relationship between the categorization of each electrode and the perceptual qualities evoked on the electrode. Contingency tables were developed for each descriptor and responses were row-divided by the three categories (LFP, IFP, and HFP) and column-divided by the presence or absence of the quality. Each category was compared pairwise. Fisher's exact test was used instead of a chi-squared test because the sample sizes for each group were relatively small.

To test whether there was spatial clustering of the effects of frequency on perceived intensity across the array, we adopted a technique used in geographic information systems, where they are described as LISA (*Anselin, 1995*). We quantified the number of electrodes that had an adjacent

electrode with the same frequency response category and divided this by the total number of adjacent electrodes to obtain a fraction. We then randomly distributed the categorized electrodes on two simulated arrays with the same tested electrode locations. We conducted this simulation 100,000 times and compared the output values of this random simulation to the observed values. A pseudo p-value was obtained by comparing the total number of simulations that had a fraction greater than or equal to the observed fraction, which indicates the probability of obtaining our observed value by chance.

For all statistical tests, we considered p<0.05 to be significant.

## Data and code availability

Data and code for this paper are available at GitHub (https://github.com/chughes003r/FrequencyPaper, *Hughes et al., 2021b*; copy archived at swh:1:rev:96f81aa826f68b9f509a3d73b7765a68ce0193e4).

## Acknowledgements

We would like to acknowledge N Copeland and Mr. Dom for their extraordinary commitment to this study and insightful discussions with the study team, as well as Debbie Harrington (Physical Medicine and Rehabilitation) for regulatory management of the study. This work was supported by the Defense Advanced Research Projects Agency (DARPA) and Space and Naval Warfare Systems Center Pacific (SSC Pacific) under Contract N66001-16-C4051 and the National Institute of Neurological Disorders and Stroke of the National Institutes of Health under Award Numbers UH3NS107714 and U01NS108922. SNF was supported by an NSF Graduate Research Fellowship under grant number DGE-1247842. Any opinions, findings, and conclusions or recommendations expressed here are those of the authors and do not necessarily reflect the views of DARPA, SSC Pacific, or the National Institutes of Health. The funders had no role in the study design, data collection, interpretation of the results, or the decision to submit this work for publication.

## Additional information

### Funding

| Funder | Grant reference number | Author |
|---|---|---|
| Defense Advanced Research Projects Agency | N66001-16-C4051 | Michael Boninger<br>Jennifer L Collinger<br>Robert A Gaunt |
| National Institutes of Health | UH3NS107714 | Michael Boninger<br>Jennifer L Collinger<br>Robert A Gaunt |
| National Institutes of Health | U01NS108922 | Michael Boninger<br>Jennifer L Collinger<br>Robert A Gaunt |
| National Science Foundation | DGE-1247842 | Sharlene N Flesher |

The funders had no role in study design, data collection and interpretation, or the decision to submit the work for publication.

### Author contributions

Christopher L Hughes, Conceptualization, Data curation, Software, Formal analysis, Validation, Investigation, Visualization, Methodology, Writing - original draft, Writing - review and editing; Sharlene N Flesher, Conceptualization, Data curation, Software, Investigation, Visualization, Methodology, Writing - review and editing; Jeffrey M Weiss, Data curation, Software, Methodology, Writing - review and editing; Michael Boninger, Resources, Funding acquisition, Project administration, Writing - review and editing; Jennifer L Collinger, Resources, Supervision, Funding acquisition, Project administration, Writing - review and editing; Robert A Gaunt, Conceptualization, Supervision, Funding acquisition, Validation, Methodology, Project administration, Writing - review and editing

## Author ORCIDs
Christopher L Hughes (iD) https://orcid.org/0000-0001-9257-8659
Jeffrey M Weiss (iD) https://orcid.org/0000-0003-1332-674X
Michael Boninger (iD) https://orcid.org/0000-0001-6966-919X
Jennifer L Collinger (iD) https://orcid.org/0000-0002-4517-5395
Robert A Gaunt (iD) https://orcid.org/0000-0001-6202-5818

## Ethics
Clinical trial registration ClinicalTrials.gov (NCT0189-4802).

Human subjects: This study was conducted under an Investigational Device Exemption from the U.S. Food and Drug administration, approved by the Institutional Review Boards at the University of Pittsburgh (Pittsburgh, PA) and the Space and Naval Warfare Systems Center Pacific (San Diego, CA), and registered at ClinicalTrials.gov (NCT0189-4802). Informed consent was obtained before any study procedures were conducted.

## Decision letter and Author response
Decision letter https://doi.org/10.7554/eLife.65128.sa1
Author response https://doi.org/10.7554/eLife.65128.sa2

# Additional files

## Supplementary files
• Transparent reporting form

## Data availability
Data and code for this paper are available at GitHub (https://github.com/chughes003r/FrequencyPaper, copy archived at https://archive.softwareheritage.org/swh:1:rev:96f81aa826f68b9f509a3d73b7765a68ce0193e4).

The following dataset was generated:

| Author(s) | Year | Dataset title | Dataset URL | Database and Identifier |
|---|---|---|---|---|
| Hughes CL, Flesher SN, Weiss JM, Boninger M, Collinger JL, Gaunt RA | 2021 | Code and data for "Perception of microstimulation frequency in human somatosensory cortex" | https://github.com/chughes003r/Frequency-Paper | GitHub, github.com/chughes003r/FrequencyPaper |

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
