## [Decision Letter]

**Acceptance summary:**

This paper characterizes percepts evoked by micro-stimulating the human somatosensory cortex. The study provides new insight into the organization of the human somatosensory cortex and represents an important step in providing more effective somatosensory feedback for brain-machine interface users.

**Decision letter after peer review:**

Thank you for submitting your article "Perceptual responses to microstimulation frequency are spatially clustered in human somatosensory cortex" for consideration by *eLife*. Your article has been reviewed by 4 peer reviewers, and the evaluation has been overseen by Andrew Pruszynski as the Reviewing Editor and Barbara Shinn-Cunningham as the Senior Editor. The following individuals involved in review of your submission have agreed to reveal their identity: Tobias Heed (Reviewer #2); Silvestro Micera (Reviewer #3).

Essential revisions:

1) The observed spatial clustering requires more evidence to accept it as a physiologically significant finding. To ultimately do so, the authors likely need additional analyses and more data (see specific comments by Reviewers #1, 2, 4). While more analysis can help note that we are not suggesting more data is needed for this paper. However, this finding needs to be toned down and approached more cautiously to avoid over-interpretation. Changing the title needs to be a part of such changes.

2) The authors should provide a more cohesive high level take-away message from the study. Although all reviewers felt the work was important and exciting from a technical perspective, the paper lacks a strong cohesive message that synthesizes the results either for BMIs or for S1 organization.

3) As laid out by Reviewer #4, it is important that the authors place their efforts in context of previous work that systematically examined effects of related stimulus parameters in human somatosensory thalamus. In general, several reviewers provide very useful literature that should be cited appropriately and discussed as needed.

*Reviewer #1 (Recommendations for the authors):*

Thank you for a really interesting manuscript.

I suggest to have a closer look at Prsa et al. Nature 2019 which shows stimulus-response curves for cortical neurons in mice showing similar dependency of stimulation frequency as observed in the current study.

Title: I am not convinced at all that there are spatially organised clusters as the title suggests. The number of electrodes is too small and there are many factors (including technical) which might explain why the electrode properties are not at random. I would be very careful making such a statement.

The electrodes are spaced 400 μm apart, but is it known within how large an area around the electrode neurons are activated?

Figures: All figures report SEMs, however SDs or CI would be more appropriate to evaluate variability.

Lines 149-153: I don't see contradiction here as HFP electrodes in general had very low median intensity. Figure 2A also shows that for HFP electrode at 20 μm the increased frequency leads to more intense percept.

Line 167: see Callier et al. PNAS 2020 who developed paradigms to address this.

Paragraph 208-225: The logic and writing is not very clear. Can you please clarify your views and refine the text?

217-219: Do you mean that convergent inputs from different sub-modalities would necessarily imply uniformity of distribution? This is not what studies mentioned in that context would suggest. Please clarify this statement.

Line 225: consider ref Birznieks et al. *eLife* 2019.

Paragraph 226-233: consider reference to Prsa et al. Nature 2019.

Lines 235-241: out of context it is not clear what are distal neurons.

Section 258-286: I disagree that there is a contradiction with non-human primate studies as I indicated in the public review. To play devil's advocate, the term "intensity" could be used to describe stimuli of different qualities and even modalities, but does it mean that, for example, intensity of olfactory stimuli could be compared with intensity of pain or any other modality? What about intensity of skin stretch vs vibrotactile intensity? Would you expect the same neural code in all those instances? Thus the sense of intensity due to electrical stimulation of cortical neurons may not be compatible with intensity of mechanical stimuli which have indentation depth, certain spatial pattern of afferent activation including size of the area, it's shape and type of borders (sharp/diffuse). All of those features may interact between each other creating a complex integrated percept of intensity which is absent with electrical stimulation.

Page 8: statements in two sections in regard to frequency filtering in cortex and human psychophysics: One possibility is that the function of some of these electrically stimulated circuits might be to detect presence of a given stimulus quality (specific qualitative feature) which might be associated with specific discharge rate in those neurons – like, for example, discharge rate <60Hz (>15ms inter-spike intervals) might mediate sense of flutter. There is experimental evidence which might support such arrangements preferentially detecting inter-spike intervals of certain length – in case of presence of short (<15ms) and long inter-spike intervals in afferent activity, the short inter-spike intervals (corresponding to high frequency) are simply ignored and perceived frequency corresponds exclusively to the longest (flutter range) inter-spike interval and not periodicity or mean discharge rate. When shorter inter-spike intervals became longer, their weighted contribution to perceived frequency increases. This has been demonstrated with mechanical (Birznieks and Vickery Current Biology 2017) and electrical stimulation (Ng et al. PlosOne 2020) and might fit with the current observations. Vibrotactile stimuli are very simplistic, but they can demonstrate principles how neural activity translates into perception.

Methods section: Please mention where the reference electrode was.

*Reviewer #2 (Recommendations for the authors):*

I am wary about the fitting and clustering approaches.

– For instance, in Figure 1, the purple curve fit of panel A seems inappropriate – the data would be better fit with a monotonically decreasing function.

– Similarly, the data in C seem to differ from D only in the very first data point of each data series. So the difference between C and D appears to be at a very low frequency only (?). I cannot assess whether this is generally true as only 2 curves are illustrated, and the use of 100 Hz in the other parts of the manuscript suggest otherwise. As is, this is an unclear point for me.

– If there is actually more spread of the preferred frequency, then I wonder how adequate it is to cluster, rather than assume, say, a continuously changing gradient across a larger cortical region.

– Related, I understand that 3 comes out as adequate from the clustering analysis, but this is based on a small number of sites. This is inherent in using electrode arrays. If more sites were available, do the authors think a 3 cluster interpretation would still be supported? What speaks against a more fine-grained, e.g. gradient-like, distribution? In other words, is there evidence against such an organization, or is the 3 cluster solution possibly just due to the small amount of testable regions/electrodes? Do the authors prefer a categorical account because they think there is a direct link to distinct perceptual qualities?

– I find it difficult to take away a "higher-order" result. To me, the presented work is clearly illuminating in that the different tested manipulations must be acquired to know in what kind of percepts they result. Also, the finding that supra-threshold stimulation results in different conclusions than near-threshold stimulation seems to me an important point. However, in many passages, the paper reads mainly like a report of all the different detailed tested aspects. I am missing some more "visionary" ideas about what the results might mean.

*Reviewer #3 (Recommendations for the authors):*

The paper is interesting and well-written. I think it could be improved by addressing the following issues:

1. The interplay between frequency and amplitude is of course very important: it could be interesting to see whether the authors could find a unifying model/equation as done in Graczyk et al., 2016 for PNS stimulation

2. Figure 1 seems to show that many electrodes have a small "intensity dynamics": could this be a problem for closed-loop control? The authors should elaborate a bit more on this

3. Figure 4 is very interesting but it could stronger and more convincing by testing more electrodes

4. the last section of the discussion on personalization is very interesting and the authors should elaborate a bit more also in this case. What kind of solution do they have in mind? Biomimetic? or Machine learning based?

5. it is not clear the duration of the overall testing. This is important to gather more information about the stability of the sensations over time.

*Reviewer #4 (Recommendations for the authors):*

A few questions and suggestions:

The major one being that the authors need to put their work in context of the original work that systematically examined effects of stimulus parameters in human somatosensory thalamus (Dostrovsky et al. 1993 Adv Neurol). Of course more publications exist on cortex recently but several of the concepts presented here as new were already studied and understood to be true in human thalamus. Also the major limitation is that all these experiments were performed in deafferented cortex; it is not just that it is one subject, but that their cortex must be reorganized. Of course that is what makes the prosthetic side of the story stronger (rather than the comparisons to normal non-human primate experiments), in that it remains amazing that sensations can be elicited from cortex that has not received afferent input in years.

1. To improve the wording of second last sentence in abstract. It does summarize the results but is somewhat confusing: is it the electrodes that were of 3 types or the brain sites stimulated.

2. While I understand how difficult it is to plot the different sensory modalities on a graph and I appreciated the radar plots in Figure 3 as the best way to do this, I was wondering how much altering different frequencies and intensities could alter the percepts evoked through a single microelectrode. Because it is unlikely we will have multiple opportunities to move cortical arrays in humans, to make this a practical application we need to know how much we can use electrical parameters to modulate percepts.

3. I did not follow the pseudo-p LISA statistical analysis shown in Figure 4. I see what the authors are trying to say in Figure 4A, but Figure B may either be simplified, better explained in the legend or perhaps moved to supplemental.

4. Why did the authors not test the effects of pulse width. It appears they were trying to test the entire parameter space that can be applied with electrical microstimulation, why not this one?

5. What about applying stimulation through multiple electrodes simultaneously? In fact this may help answer the question about whether short-term plasticity is involved (from discussion).

6. Please clarify if this patient is the same one reported in previous papers. The introduction suggested that it was and perhaps this was why the projected fields evoked with microstimulation were not described. If this is the case then how did Figure 4 compare to the somatotopy described in the previous paper? And what part of S1 do they believe the array is located (i.e. what Brodmann area 1, 2, 3a,b)? Adding a statement about what cortical layer the authors believe the microelectrodes are located before the limitations section of the discussion would be helpful. There are a few words about somatotopy at the end of the Results section indicating that all three types of responses: low, med, and high frequency preferring regions can subserve the same body region. However because the statement says "in some cases", it makes it equivocal, yet is a major component of the discussion.

7. The authors describe that because the subject is deafferented they could not identify receptive fields from the recordings. Were recordings performed at all in the deafferented cortex? While that is not the subject of this paper, it would be a welcome addition to the literature.

8. Please clarify the consistency of results longitudinally. Figure S3 does not seem to demonstrate all electrode showing same results over time: 3 of 7 electrodes (2, 3, 36) seem different over time.

---

## [Author Response]

Essential revisions:1) The observed spatial clustering requires more evidence to accept it as a physiologically significant finding. To ultimately do so, the authors likely need additional analyses and more data (see specific comments by Reviewers #1, 2, 4). While more analysis can help note that we are not suggesting more data is needed for this paper. However, this finding needs to be toned down and approached more cautiously to avoid over-interpretation. Changing the title needs to be a part of such changes.

Thank you for these helpful comments. We will address the individual concerns throughout this document relevant to each of these points. However, here we summarize how we plan to address these essential revisions.

Based on the robust statistical testing, we are confident that the responses were not randomly distributed across the cortex in these experiments and that they do indeed cluster. However, the analysis itself may need to be explained better. We have addressed this point later in the response and in the main text. However, we agree that with just a single participant and without more extensive datasets, making strong claims about structural organization may not be appropriate. However, since receiving the responses we have collected a smaller data set in a second participant and observed similar clustering which is now included as figure supplements (Figure 1—figure supplement 4, Figure 4—figure supplement 1). Information about these data have been added to the manuscript. While we believe this demonstrates even more strongly that these relationships are spatially clustered, the reviewers are correct to note there are other possible explanations for this observed clustering. Alternative explanations, including layer specificity and more continuous variation in these relationships, were considered, but ultimately removed from the original paper to make discussion more concise. These points have been added back into the discussion. Nevertheless, we will also deemphasize spatial clustering as a major point of the paper by changing the title as follows:

Title: Perception of microstimulation frequency in human somatosensory cortex

2) The authors should provide a more cohesive high level take-away message from the study. Although all reviewers felt the work was important and exciting from a technical perspective, the paper lacks a strong cohesive message that synthesizes the results either for BMIs or for S1 organization.

This study has important implications for both BCI development and for basic studies of human brain structure and function. Based on point 1 above, we will de-emphasize claims related to the latter point, although we still feel that it is important to discuss these. The critical conclusions of this study are, (1) stimulation frequency does not have consistent linear effects on percept intensity across electrodes as we have seen with amplitude, (2) stimulation frequency changes the perceptual qualities in different ways on different electrodes, (3) these intensity and quality effects are coupled, and (4) bidirectional BCIs will need to consider these electrode-specific effects to implement effective feedback algorithms. Importantly, these disparate effects of frequency on perception were not predicted from prior animal work and have implications for the structure and function of the brain that need to be studied further. The revised manuscript now emphasizes these points clearly to present a cohesive story.

3) As laid out by Reviewer #4, it is important that the authors place their efforts in context of previous work that systematically examined effects of related stimulus parameters in human somatosensory thalamus. In general, several reviewers provide very useful literature that should be cited appropriately and discussed as needed.

We would like to thank the reviewers for pointing out the literature on thalamic stimulation. This is an important body of work we have discussed in other work. However, we originally chose not to reference this here because it is likely that the effects we observed are limited to cortical stimulation. However, we will reference this work and draw links to the cortical stimulation experiments that we conducted.

Reviewer #1 (Recommendations for the authors):Thank you for a really interesting manuscript.I suggest to have a closer look at Prsa et al. Nature 2019 which shows stimulus-response curves for cortical neurons in mice showing similar dependency of stimulation frequency as observed in the current study.

Thank for the positive feedback. Prsa et al. 2019 is indeed a very relevant paper and we find the parallels between our own work and the Prsa study very intriguing. Specifically they found that different neurons in S1 responded very differently to mechanical stimulation at different frequencies (similar to our finding that different ICMS frequencies evokes different perceptual and intensity effects). They also found that decreasing the mechanical stimulus amplitude biased the preferred frequency higher (conceptually similar to our finding that as ICMS current decreases, higher stimulation frequencies are more easily detected). However, it is important to note that there are several key differences between the studies that complicate any direct comparison. Specifically, the most substantial effects in our study occurred in a frequency range from 20-100 Hz, which incorporates the ‘flutter’ range where cortical neuron firing is phase-locked to the stimuli. Conversely, Prsa studied frequencies from 100-800 Hz where phase-locking does not occur. The following text has been added to the discussion (Line 243):

“Similarly, individual cells in mouse cortex are preferentially activated by different mechanical stimulation frequencies (Prsa et al., 2019). In those same experiments, the frequency preference of the neural population tended towards higher frequencies when the indentation depth decreased, similar to our results that higher frequencies were perceived as being more intense when the ICMS amplitude was decreased (Figure 2).”

Title: I am not convinced at all that there are spatially organised clusters as the title suggests. The number of electrodes is too small and there are many factors (including technical) which might explain why the electrode properties are not at random. I would be very careful making such a statement.

Thank you for the feedback. This was a common concern amongst reviewers. We agree that the number of samples on each array was small, however we are confident that the statistical analysis support the idea that the responses were clustered in these experiments. However, as you and others noted, this could be due to a variety of factors that were not previously emphasized, including layer specificity and potentially electrode-tissue interactions. Further, with data from a single subject, it remains to be shown whether such a finding is broadly true. However, we now have data from a second participant in which similar clustering was observed. We have updated the discussion to emphasize alternative explanations and the point that while the relationships are clustered, this may or may not be due to cortical organization. Further, we have revised the title to not include this issue about spatial clustering.

The title is now:

“Perception of microstimulation frequency in human somatosensory cortex”.

The following text was added to the discussion (Line 328):

“Our results are consistent with the idea that somatosensory cortex is organized in a way that represents different features in different locations, however, there are several limitations that should be considered. […] However, if this were the case, this would still reflect important functional differences in cortex which need to be understood for bidirectional BCIs.”

The electrodes are spaced 400 μm apart, but is it known within how large an area around the electrode neurons are activated?

This is a great question that unfortunately doesn’t have a simple answer. While there is literature measuring and simulating how stimulation activates elements as a function of stimulus parameters and distance from the electrode (see (Overstreet et al., 2013; Stoney et al., 1968)), this can depend on several factors that are difficult to measure including any tissue encapsulation. Activation volumes have been estimated to be anywhere from a few hundred microns to a few millimeters for 100 μA currents (Overstreet et al., 2013). Perhaps the more complicated issue is that local stimulation can activate passing axons that project to distant cortical areas. Studies in mice have shown that stimulation can sparsely activate distant neurons hundreds of microns from the stimulating electrode (Histed et al., 2009; Michelson et al., 2019) but this itself may be frequency dependent. Ultimately, it is difficult for us to know exactly what volume of tissue is activated by ICMS. Nevertheless, passing axons close to electrodes may themselves be organized (by layer or column) and could contribute to clustering we observed. With our setup, the precise neural mechanisms that result in “frequency preference” is elusive, and will likely require further studies in animal models. We have added the following text to the discussion about the effects of ICMS on local and distant neural populations (Line 216):

“Intracortical microstimulation at the maximum amplitudes used in this study can directly activate tissue up to 2 mm away from the electrode tip, but most activation occurs less than 500 µm from the electrode tip (Overstreet et al., 2013; Stoney et al., 1968). […] These spatial scales over which function varies are similar to the expected recruitment distances from ICMS, supporting the idea that different perceptual or frequency responses may be due to activation of different functional groups of neurons.”

Figures: All figures report SEMs, however SDs or CI would be more appropriate to evaluate variability.

We agree that SD or CI would be most appropriate for assessing variability within a group. However, here, we are primarily interested in whether the mean values at different frequencies are different for the three different categories of electrodes. In this case, standard error shows how well we are estimating the mean, and we feel this is the most appropriate metric. Psychophysical studies of electrical stimulation typically use SEM in similar situations (Callier et al., 2019; Graczyk et al., 2016; Kim et al., 2015b, 2015a). Standard deviation would instead be informative about how much the participants’ intensity reports varied on different electrodes within a group. While the overall magnitude of the intensity is related to the group with which an electrode is divided into, the specific numbers used can vary from one electrode to another and even from day to day (although we found the day-to-day variance was not significant). How the intensity measures vary on electrodes within a group is not of interest and the standard deviation then doesn’t really illustrate what we are interested in. Therefore, we believe that the standard error is the more appropriate statistical metric to report.

Lines 149-153: I don't see contradiction here as HFP electrodes in general had very low median intensity. Figure 2A also shows that for HFP electrode at 20 μm the increased frequency leads to more intense percept.

Thank you for this comment. We have revised this statement to try and better capture our thoughts. Briefly, we don’t believe that our data are contradictory with the findings in NHPs. Rather, the predictions that were made from NHPs (included in the discussion of both Callier et al. 2020 – see Discussion section titled “Increased ICMS Frequency Leads to Increased Perceived Magnitude” – and Kim et al. 2015 –see Discussion section titled “Effects of Frequency.”) may be inaccurate. We work closely with the group responsible for this work (the Bensmaia group) and have spoken at length about how our results work together. We believe that the detection results from the human participant confirm that the directly-comparable data (detection thresholds) are not contradictory as the results from stimulation at 20 µA (Figure 2) is more in line with what is expected based on the primate work. However, the frequency effects we saw at higher amplitudes could not be predicted from monkeys because these percepts would always be detectable and monkeys can’t reasonably report intensity. Therefore, we believe these results do not contradict NHP findings, but rather give us new insight that the monkey experiments could not. To clarify our thoughts, here are the updated passages (Line 160):

“Our observation that higher stimulus pulse frequencies can evoke less intense percepts at suprathreshold stimulus current amplitudes differs from predictions made from non-human primate studies; higher frequencies evoked detectable percepts at lower amplitudes in NHPs, which led to predictions that higher frequency always results in higher perceived intensities ((S. Kim, Callier, Tabot, Gaunt, et al., 2015; Romo et al., 2000, 1998).”

The remainder of this paragraph presents the idea that increases in perceived intensity only occur consistently when the stimulus amplitudes are very low. It is only at higher amplitudes where the differences begin to emerge (Figure 2).

Additionally, we changed the title of the Discussion section related to this point (Line 275):

“ICMS in humans directly evaluates intensity and perception”.

This revised Discussion section expands upon this point.

Line 167: see Callier et al. PNAS 2020 who developed paradigms to address this.

In this paper, the authors vary amplitude and frequency together to separate out the effects of amplitude and frequency on perception. They found that amplitude biases discrimination of frequency on some electrodes more than others. And still on other electrodes, the monkeys are not able to perform the task. We believe that non-linear effects of frequency on perception could result in monkeys being unable to perform the discrimination task. The following text in the Discussion directly addresses this point (Line 284):

“To determine if changes in frequency could be perceived independently of changes in amplitude, animals were trained to identify which of two intervals contained the higher frequency stimulus train, regardless of current amplitude (Callier et al., 2020). […] On electrodes without highly salient frequency-dependent qualities, such as the HFP electrodes, it would be difficult to disambiguate changes in amplitude and frequency.”

Paragraph 208-225: The logic and writing is not very clear. Can you please clarify your views and refine the text?

We apologize for the confusion that this text created. Briefly, the major point of this text is that previous literature emphasized the importance that specific types of peripheral afferents (RAs, SAs, PCs), had on the response in cortex. More specifically, the thought was that cortical neurons basically just reflected the way that different peripheral afferents responded to mechanical input. If this were true, direct cortical input through ICMS should have had the same effects everywhere. Since we saw that there were differences in intensity and percept quality at the same frequencies in different locations, this suggests that different regions of the cortex may be tuned to process ‘features’ of the input, and not simply reflect peripheral afferent activity. This is consistent with recent work in the mouse (Prsa et al., 2019)Click or tap here to enter text. and the fact that different peripheral submodaliti es (SAs, RAs, PCs) actually all converge on many cortical neurons (Pei et al., 2009). We have significantly revised this section and hope that it is now more clear (Line 228):

“Electrophysiological (Mountcastle, Talbot, Sakata, and Hyvärinen, 1969; Sur, Wall, and Kaas, 1981; Sur et al., 1984) and optical (L. M. Chen, Friedman, Ramsden, LaMotte, and Roe, 2001; Friedman et al., 2004) recordings have shown organized neural populations in the somatosensory cortex that are sensitive to tactile input with specific frequency content. […] Therefore, the different effects of stimulus frequency on intensity and perception are unlikely to arise from activating inputs representing specific tactile subpopulations (e.g. SAs, RAs, or PCs), but by how a local region of somatosensory cortex can respond to different stimulus frequencies, consistent with the concept of the cortex encoding different frequency features (Prsa, Morandell, Cuenu, and Huber, 2019).”

217-219: Do you mean that convergent inputs from different sub-modalities would necessarily imply uniformity of distribution? This is not what studies mentioned in that context would suggest. Please clarify this statement.

We apologize that this statement did not accurately convey our thoughts. We did not mean to imply that all cortical neurons receive equal input from all tactile submodalities. In the dorsal column nuclei and thalamus there is a clearer preservation of submodality specific activity. However, the study by Pei (Pei et al., 2009)Click or tap here to enter text. shows that this concept breaks down in the cortex with many cortical neurons representing activity that is associated with both rapidly and slowly adapting responses. In area 1 (where our arrays are implanted), about 40% of the neurons received these mixed inputs. The papers we referenced were presented as a counterpoint to papers (e.g. (Friedman et al., 2004; Sur et al., 1984)) that suggest that cortical neurons can be classified simply as SA or RA neurons. We believe that our work provides further evidence that the somatosensory cortex does not appear to be organized by segregated input modalities consistent with labelled line inputs, but rather the cortex itself is organized in a manner to preferentially encode different features of perception. This text has been revised as follows (Line 231):

“These experiments promoted the idea of submodality separation in the cortex in which the activity of cortical neurons is primarily driven by input from either rapidly adapting Meissner corpuscles (RAs), slowly adapting Merkel cells (SAs), or Pacinian Corpuscles (PCs). However, many cortical neurons receive heterogeneous input from multiple classes of mechanoreceptors (Pei, Denchev, Hsiao, Craig, and Bensmaia, 2009; Reed et al., 2010; Saal and Bensmaia, 2014), leading to neurons that exhibit both sustained and transient responses.”

Line 225: consider ref Birznieks et al. eLife 2019.

Thanks for the suggestion. We have added the following sentence to our discussion (Line 241):

“The idea that somatosensory cortex is organized for feature encoding is supported by human psychophysics experiments where frequency perception was dependent on specific spiking patterns and not on the types of mechanoreceptor that were activated (Birznieks et al., 2019).”

Paragraph 226-233: consider reference to Prsa et al. Nature 2019.

Thanks for this suggestion. This paragraph was revised as follows (Line 243):

“Similarly, individual cells in mouse cortex are preferentially activated by different mechanical stimulation frequencies (Prsa et al., 2019). In those same experiments, the frequency preference of the neural population tended towards higher frequencies when the indentation depth decreased, similar to our results that higher frequencies were perceived as being more intense when the ICMS amplitude was decreased (Figure 2).”

Lines 235-241: out of context it is not clear what are distal neurons.

We agree that this term may have been unclear and have revised this paragraph to use a more accurate and descriptive phrase. Essentially, we meant “neurons that are far away from the electrode”. The revised paragraph now reads (Line 252):

“The effects described here must be related to different cellular responses to stimulation in different regions of the cortex. […] This reduced activity in neurons far away from the electrode could lead to decreases in perceived intensity and changes in percept quality in a way that depends on electrode location and local neural populations.”

Section 258-286: I disagree that there is a contradiction with non-human primate studies as I indicated in the public review. To play devil's advocate, the term "intensity" could be used to describe stimuli of different qualities and even modalities, but does it mean that, for example, intensity of olfactory stimuli could be compared with intensity of pain or any other modality? What about intensity of skin stretch vs vibrotactile intensity? Would you expect the same neural code in all those instances? Thus the sense of intensity due to electrical stimulation of cortical neurons may not be compatible with intensity of mechanical stimuli which have indentation depth, certain spatial pattern of afferent activation including size of the area, it's shape and type of borders (sharp/diffuse). All of those features may interact between each other creating a complex integrated percept of intensity which is absent with electrical stimulation.

We have modified this section to be clearer about our thoughts and think that our use of the word ‘contradicts’ was misleading. We have therefore removed this term and focus on the idea that our data appear to be inconsistent with the *predictions* about the effects of frequency on intensity that have been made from non-human primate studies. With good reason, several papers suggested that increasing ICMS frequency in somatosensory cortex will always result in increases in perceived intensity (see discussions from (Callier et al., 2019; Kim et al., 2015b)). However, in our results we see changes in perceived intensity that are clearly different from these predictions. There are several factors that could lead to these differences. First, in this study we directly measure perceived intensity whereas this can only be inferred indirectly in monkeys. Second, there was a relationship between the way that ICMS frequency modulated intensity and percept quality. When the quality changes, it is possible that certain percepts are under associated with lower intensities. However, there is literature to support the idea that cross-modal intensity matching can in fact be done (Marks et al., 1986). Nevertheless we more clearly point out that intensity and quality both change as frequency changes and specifically address the possibility that quality changes could be a factor in intensity perception. Abstract, Line 27:

“These results support the idea that stimulation frequency directly controls tactile perception and that these different percepts may be related to the organization of somatosensory cortex, which will facilitate principled development of stimulation strategies for bidirectional BCIs.”

Discussion, Line 299:

“Another interpretation is that since frequency can change percept quality (Figure 3), different qualities are understood to have different intensities.”

Page 8: statements in two sections in regard to frequency filtering in cortex and human psychophysics: One possibility is that the function of some of these electrically stimulated circuits might be to detect presence of a given stimulus quality (specific qualitative feature) which might be associated with specific discharge rate in those neurons – like, for example, discharge rate <60Hz (>15ms inter-spike intervals) might mediate sense of flutter. There is experimental evidence which might support such arrangements preferentially detecting inter-spike intervals of certain length – in case of presence of short (<15ms) and long inter-spike intervals in afferent activity, the short inter-spike intervals (corresponding to high frequency) are simply ignored and perceived frequency corresponds exclusively to the longest (flutter range) inter-spike interval and not periodicity or mean discharge rate. When shorter inter-spike intervals became longer, their weighted contribution to perceived frequency increases. This has been demonstrated with mechanical (Birznieks and Vickery Current Biology 2017) and electrical stimulation (Ng et al. PlosOne 2020) and might fit with the current observations. Vibrotactile stimuli are very simplistic, but they can demonstrate principles how neural activity translates into perception.

Thanks for the interesting thoughts. We certainly agree that the observed frequency responses might be a consequence of detecting specific qualitative features. This was a point we tried to make in the discussion and make more clearly now (see below). We also agree that somatosensory cortex may encode certain ‘features’, and may be more sensitive to large interpulse spacings. These could be the electrodes we identified as “low” or “intermediate” preferring. This would imply that the other areas of cortex, the “high frequency preferring” are sensitive not to the interpulse spacing, but rather the overall rate (or number of pulses). The results from the Birznieks lab are definitely relevant to our work. In fact, we have conducted another set of experiments in which we observed if the changes in the interpulse timings affected perception of ICMS in a way that was consistent with the Birznieks papers cited here. For the small data set collected, we found this to be true: increasing pulse spacing resulted in a decrease in perceived frequency. This was presented as a 4-page paper at the IEEE NER conference titled “Changes in interpulse spacing changes perception of microstimulation in human somatosensory cortex.”

Line 241:

“The idea that somatosensory cortex is organized for feature encoding is supported by human psychophysics experiments where frequency perception was dependent on specific spiking patterns and not on the types of mechanoreceptor that were activated (Birznieks et al., 2019).”

Methods section: Please mention where the reference electrode was.

Apologies for this omission. The ‘return’ electrode or ‘anode’ for stimulation is actually the titanium pedestal that is screwed to the skull. As a result, ICMS is effectively monopolar (distant return electrode). We have added this detail to the methods (Line 389):

“The stimulation return electrode was the titanium pedestal that was fixed to the skull.”

Reviewer #2 (Recommendations for the authors):I am wary about the fitting and clustering approaches.– For instance, in Figure 1, the purple curve fit of panel A seems inappropriate – the data would be better fit with a monotonically decreasing function.

We agree that the purple curve here should not be used to make assumptions about how these electrodes behave. We intended that the curves be understood simply for illustrative purposes to emphasize the differences between the group trends. However, given the potential confusion and the point raised here, we have revised all the figures to use simple piecewise fits between each data point. In the specific case mentioned (purple data in Figure 1A), there actually is a low intensity data point at the lowest frequency, which would make a monotonically decreasing function inappropriate. We believe that replacing these curve fits with simple connecting lines will help illustrate this more clearly and reduce confusion. Additionally, to the figure legends we have added:

“The points are connected with piecewise fits.”

– Similarly, the data in C seem to differ from D only in the very first data point of each data series. So the difference between C and D appears to be at a very low frequency only (?). I cannot assess whether this is generally true as only 2 curves are illustrated, and the use of 100 Hz in the other parts of the manuscript suggest otherwise. As is, this is an unclear point for me.

We agree that most of the changes appear in the lowest frequency ranges. Above about 150 Hz, there were only small relative changes in intensity within a group of electrodes. However, in Figure 1 B,C,D, we did not keep the y-axis range consistent in order to emphasize the shape of the relationships on individual electrodes, which may lead to some misinterpretation. Panel A shows the dominant effects more clearly. The low and intermediate groups have similar intensity responses at 20 Hz, but the intermediate group has higher intensities at all other frequencies. Specifically, both low and intermediate groups have intensities of 1.5-2 around 20 Hz. Low groups then have intensities of 1-1.5 for 40+ Hz and intermediate groups have intensities of 2-2.5 for 40+ Hz. This is what results in the different shapes of the curves.

– If there is actually more spread of the preferred frequency, then I wonder how adequate it is to cluster, rather than assume, say, a continuously changing gradient across a larger cortical region.

This is an interesting suggestion, and we think that it is possible that these effects could be more continuous rather than being divided into the three groups as we proposed here. To determine these groups we used both silhouette and elbow analysis of our k-means clustering results to confirm that separating into three groups was reasonable. And in fact, in our second participant, clustering into two groups was more reasonable based on this same approach. However, with additional data from new electrodes distributed across larger regions of the cortex, it is certainly possible that more continuous distributions could emerge. In our specific case however, three groups captured the data well. We have added the following text to the discussion to address this point (Line 328):

“Our results are consistent with the idea that somatosensory cortex is organized in a way that represents different features in different locations, however, there are several limitations that should be considered. […] However, if this were the case, this would still reflect important functional differences in cortex which need to be understood for bidirectional BCIs.”

– Related, I understand that 3 comes out as adequate from the clustering analysis, but this is based on a small number of sites. This is inherent in using electrode arrays. If more sites were available, do the authors think a 3 cluster interpretation would still be supported? What speaks against a more fine-grained, e.g. gradient-like, distribution? In other words, is there evidence against such an organization, or is the 3 cluster solution possibly just due to the small amount of testable regions/electrodes? Do the authors prefer a categorical account because they think there is a direct link to distinct perceptual qualities?

Thank you for these interesting comments. This response is related to several of the previous responses. We believe that it is possible that the divisions that we described are in fact more continuous. We simply do not have the data to conclusively rule out either of these two interpretations. However, the three groups were parsimonious explanations of the data and there are physiological reasons to think that the responses may be more effectively considered to divide into groups, rather than be distributed across a continuum. While it might be tempting to think that the three divisions align with the frequency response ranges of SA1, RA and PC mechanoreceptors, this had nothing to do with how we approached it. Neurons in somatosensory cortex can receive converging input from different classes of mechanoreceptors, and as such, these frequency response ranges probably do not directly reflect mechanoreceptor responses. An alternative view from optical imaging experiments showed these relationships as being more continuous (Friedman et al. 2004). Nevertheless, the categorical account given here made sense from our observation and clustering analysis, particularly when considering the quality reports. We have added the following text to the *Limitations* section of the Discussion to address this issue (same text as previous question and response).

– I find it difficult to take away a "higher-order" result. To me, the presented work is clearly illuminating in that the different tested manipulations must be acquired to know in what kind of percepts they result. Also, the finding that supra-threshold stimulation results in different conclusions than near-threshold stimulation seems to me an important point. However, in many passages, the paper reads mainly like a report of all the different detailed tested aspects. I am missing some more "visionary" ideas about what the results might mean.

We hope that this revised manuscript more clearly addresses this point and we apologize that our broader vision of the importance of these experiments did not come through clearly. The reviewer is correct in that these data provide new information about relevant factors that must be considered to construct bidirectional brain-computer interfaces. However, more importantly, we believe that this work provides direct insight into the organization of perception in somatosensory cortex that can only be inferred from animal experiments. While we knew that different regions of the cortex could respond to different types of mechanical input, it was not clear whether this itself was causally linked to different percepts. Stimulation experiments, as we did here, provides this causal link. The text has been revised throughout, but a specific point to this effect is now the conclusion of the abstract (Abstract, Line 27):

“These results support the idea that stimulation frequency directly controls tactile perception and that these different percepts may be related to the organization of somatosensory cortex, which will facilitate principled development of stimulation strategies for bidirectional BCIs.”

Reviewer #3 (Recommendations for the authors):The paper is interesting and well-written. I think it could be improved by addressing the following issues:1. The interplay between frequency and amplitude is of course very important: it could be interesting to see whether the authors could find a unifying model/equation as done in Graczyk et al., 2016 for PNS stimulation

Thank you for this suggestion. This would be ideal. However, the key to building a function as the Graczyk paper did is the consistent effect of frequency on intensity in the periphery. Because the relationships are electrode specific here, it is not possible to develop a unifying equation that could make predictions about intensity. We could potentially create functions for each preference group, but it is not clear how useful this would be.

2. Figure 1 seems to show that many electrodes have a small "intensity dynamics": could this be a problem for closed-loop control? The authors should elaborate a bit more on this

This would be an issue if we were attempting to use frequency to modify intensity. However, because of the non-linear and electrode-to-electrode effects of frequency, it would not be a first choice to modulate intensity. Rather, modifying amplitude provides a reliable way to modulate intensity, and also allows much better dynamics (see Figure 1—figure supplement 1, although the intensity is normalized here). Amplitude allows better modulation, but we have also found multielectrode stimulation can be used for adjacent electrodes to increase intensity, and this could be another potential means to modulate intensity dynamically (but requires further investigation). Based on these results, we believe frequency could be modified based on the periodicity of the input and the electrode being stimulated, but not specifically to modulate intensity. We have tried to clarify these thoughts in the Discussion section “Implications for prostheses.” (Line 348):

“Stimulus amplitude linearly modulates intensity, while frequency has non-monotonic and electrode specific effects on intensity and percept quality. […] Future studies should assess the efficacy of these parameters.”

3. Figure 4 is very interesting but it could stronger and more convincing by testing more electrodes.

Thank you for the positive feedback. This general issue was raised by several other reviewers as well. This figure has been modified and the general results and discussion surrounding this have been changed. We agree that to fully test this basic idea more experimentation will need to be done. Please see the responses to Reviewer 1, Questions 2 and 3 for more information.

4. the last section of the discussion on personalization is very interesting and the authors should elaborate a bit more also in this case. What kind of solution do they have in mind? Biomimetic? or Machine learning based?

This is an incredibly interesting and relevant question as we are currently investigating both approaches in our lab. We are using biomimetic stimulation (based on neural recordings and TouchMime models; see references below)) to understand if biomimetic stimulation can change the perception of stimulation in a desirable way. We are also testing methods of Bayesian inference to allow parameter optimization to select desirable parameters. How these methods may be useful and how they can potentially be combined remains to be seen. We have elaborated on this in the “Implications for prostheses” section (Line 358).

“Second, these results suggest that electrode-specific stimulation encoding schemes would be particularly useful. […] These methods could ultimately improve the usefulness of somatosensory feedback, in turn improving the performance of bidirectional BCIs and ultimately improving the quality of life for people living with paralysis.”

5. it is not clear the duration of the overall testing. This is important to gather more information about the stability of the sensations over time.

We apologize for this oversight and have made this more clear in the methods section. We tried to limit all data collection to one year to avoid results being inconsistent with each other over longer periods of time. Added in Methods (Line 378):

“All data included in this paper (including magnitude estimation, surveys, detection thresholds, etc.) were limited to one year of data collection in P2 to minimize the impact of changes in perception that can occur over long time periods. Data in P3 were collected over two months.”

Reviewer #4 (Recommendations for the authors):A few questions and suggestions:The major one being that the authors need to put their work in context of the original work that systematically examined effects of stimulus parameters in human somatosensory thalamus (Dostrovsky et al. 1993 Adv Neurol). Of course more publications exist on cortex recently but several of the concepts presented here as new were already studied and understood to be true in human thalamus. Also the major limitation is that all these experiments were performed in deafferented cortex; it is not just that it is one subject, but that their cortex must be reorganized. Of course that is what makes the prosthetic side of the story stronger (rather than the comparisons to normal non-human primate experiments), in that it remains amazing that sensations can be elicited from cortex that has not received afferent input in years.

We thank the reviewer for the suggestions. We tried to find the specific article referenced here titled “Electrical stimulation-induced effects in the human thalamus” and while we were able to find the citation, we could not find the original article anywhere. We did go through all of the papers that cited this paper, of which there were 25, and looked for relevant articles. Additionally, we reviewed more recent papers on thalamic stimulation. We have attempted to include the relevant information in the introduction and discussion. We have currently added the following text to the introduction (Line 59):

“More is known about the perceptual effects of stimulating the human thalamus (Davis, Kiss, Tasker, and Dostrovsky, 1996; Heming, Sanden, and Kiss, 2010; Ohara, Weiss, and Lenz, 2004; Swan, Gasperson, Krucoff, Grill, and Turner, 2018; Willsey et al., 2020). […] Ultimately, temporal factors have clear effects on the sensations evoked through thalamic stimulation, but it remains unclear how to optimally control these parameters to manipulate percept quality.”

We agree that the cortex being deafferented is likely a major factor to consider in this work, and as such we have included more information about this in the discussion. Although much recent work has promoted the idea that cortex doesn’t actually reorganize after injury in ways previously thought which may explain why we are still able to induce naturalistic sensations years after injury. We added the following text to the Limitations section (Line 315):

“The participants had limited residual sensation in their hands, which made it difficult to measure responses in cortex to tactile indentation. […] The ability to elicit sensations with ICMS years after injury is supportive of this idea (Armenta Salas et al., 2018; Fifer et al., 2020; Flesher et al., 2016)”.

1. To improve the wording of second last sentence in abstract. It does summarize the results but is somewhat confusing: is it the electrodes that were of 3 types or the brain sites stimulated.

We apologize for the confusion. We believe that the three groups are a result of the structure and function of the different brain regions activated by stimulation. We have rewritten the end of the abstract to make this point clearer (Line 24).

“These different frequency-intensity relationships were divided into three groups which also evoked distinct percept qualities at different stimulus frequencies. […] These results support the idea that stimulation frequency directly controls tactile perception and that these different percepts may be related to the organization of somatosensory cortex, which will facilitate principled development of stimulation strategies for bidirectional BCIs.”

2. While I understand how difficult it is to plot the different sensory modalities on a graph and I appreciated the radar plots in Figure 3 as the best way to do this, I was wondering how much altering different frequencies and intensities could alter the percepts evoked through a single microelectrode. Because it is unlikely we will have multiple opportunities to move cortical arrays in humans, to make this a practical application we need to know how much we can use electrical parameters to modulate percepts.

The perceptual quality and intensity can vary significantly across frequencies for individual electrodes. The group trends are very representative of the effects on individual electrodes within the group: a low frequency preferring electrode will likely have very different percepts at 20 Hz vs 100+Hz. Stimulation amplitude on the other hand tends to only modulate the intensity and not the quality. Figure 3 summarizes the effects of changing frequency on perception. Certainly more work needs to be done in this area, but we believe that this type of analysis is representative, although it will need to be validated in more participants.

3. I did not follow the pseudo-p LISA statistical analysis shown in Figure 4. I see what the authors are trying to say in Figure 4A, but Figure B may either be simplified, better explained in the legend or perhaps moved to supplemental.

We apologize for the confusion. The intention of this figure was to demonstrate the results of the random simulations that showed the clustering measured experimentally was significantly greater than chance. However, we agreed that Figure 4B is not a critical finding that needs to be included in the main figure. In fact, we have replaced Figure 4B with the somatotopic organization across the arrays in participant P2, which we feel is a more illustrative and useful figure. The figure legend has been appropriately updated. Information about how the p-values were calculated will remain in the methods section.

4. Why did the authors not test the effects of pulse width. It appears they were trying to test the entire parameter space that can be applied with electrical microstimulation, why not this one?

This is a great suggestion, and one that we wish that we could easily manipulate. However, our clinical protocol limits pulse width because the preclinical safety studies that were performed did not vary pulse width. We expect that as new studies are conducted, and the general effects of electrical stimulation safety become more well understood, pulse width modulation could be a useful may to uniquely modulate perception.

5. What about applying stimulation through multiple electrodes simultaneously? In fact this may help answer the question about whether short-term plasticity is involved (from discussion).

This is an interesting suggestion. Multielectrode stimulation may be desirable for many reasons, and we hope to investigate using this feature in detail to understand its usefulness. We have performed limited studies showing that multielectrode stimulation can expand the dynamic range for intensity. Further, multielectrode stimulation studies are the focus of some ongoing work and we hope to share those results in the coming years.

6. Please clarify if this patient is the same one reported in previous papers. The introduction suggested that it was and perhaps this was why the projected fields evoked with microstimulation were not described. If this is the case then how did Figure 4 compare to the somatotopy described in the previous paper? And what part of S1 do they believe the array is located (i.e. what Brodmann area 1, 2, 3a,b)? Adding a statement about what cortical layer the authors believe the microelectrodes are located before the limitations section of the discussion would be helpful. There are a few words about somatotopy at the end of the Results section indicating that all three types of responses: low, med, and high frequency preferring regions can subserve the same body region. However because the statement says "in some cases", it makes it equivocal, yet is a major component of the discussion.

We apologize for the lack of clarity here. Most of the data in this manuscript are from the same participant as previous papers. We have added details to clarify this (Line 384).

“Results from this participant have been reported previously (Flesher et al., 2016, 2021; Hughes, Flesher et al., 2020).”

Because these results have been reported before, we originally chose to not report somatotopic maps. However, we have now modified Figure 4 to include this information. We believe the electrodes are in Layer IV based on the length of the electrodes, but this is impossible to confirm. It is possible that electrode arrays with multiple electrodes along the shank could more accurately determine depth, but with these electrodes, this is not possible. We have added text to the discussion for this (Line 339):

“Third, we do not know if electrodes across the array are in different layers of cortex. Different layers of cortex may drive different perceptual responses with the same input. However, if this were the case, this would still reflect important functional differences in cortex which need to be understood for bidirectional BCIs.”

The reason we did not strongly emphasize the result that the frequency preference groups could span multiple projected fields is because our sample of the cortex is very small and we really only have one example of a single projected field with more than one frequency preference. Because of this, we did not want to emphasize this as a finding of the work.

7. The authors describe that because the subject is deafferented they could not identify receptive fields from the recordings. Were recordings performed at all in the deafferented cortex? While that is not the subject of this paper, it would be a welcome addition to the literature.

Thank you for this comment. We agree that this type of information would be very useful, but as a result of the spinal cord injury, this is very difficult. We are developing a better apparatus to more explicitly probe this issue in ongoing work.

8. Please clarify the consistency of results longitudinally. Figure S3 does not seem to demonstrate all electrode showing same results over time: 3 of 7 electrodes (2, 3, 36) seem different over time.

We agree that the figure might look like this. However, there was no statistically significant change over time for any of these tested electrodes according to Friedman’s test. However, two electrodes, 3 and 36, were more variable for individual frequencies due to the low intensity of the percepts. So, while these electrodes always showed a general increase in intensity across frequencies, there was often variability in the precise shape of the curve. A contributing factor here is that each day only contains 5 trials for each tested frequency. More trials would have resulted in less variability, but would have been an issue for other reasons (e.g. participant engagement in task). This makes individual days of data contain more variability, but the statistics support that these relationships do not change significantly over time.

Callier, T., Brantly, N.W., Caravelli, A., and Bensmaia, S.J. (2019). The frequency of cortical microstimulation shapes artificial touch. P Natl Acad Sci Usa 117, 1191–1200.

Friedman, R.M., Chen, L.M., and Roe, A.W. (2004). Modality maps within primate somatosensory cortex. P Natl Acad Sci Usa 101, 12724–12729.

Graczyk, E.L., Schiefer, M.A., Saal, H.P., Delhaye, B.P., Bensmaia, S.J., and Tyler, D.J. (2016). The neural basis of perceived intensity in natural and artificial touch. Sci Transl Med 8, 362ra142 362ra142.

Histed, M.H., Bonin, V., and Reid, R.C. (2009). Direct activation of sparse, distributed populations of cortical neurons by electrical microstimulation. Neuron 63, 508 522.

Hughes, C., Herrera, A., Gaunt, R., and Collinger, J. (2020). Bidirectional brain-computer interfaces. Handb Clin Neurology 168, 163–181.

Kim, S., Callier, T., Tabot, G.A., Tenore, F.V., and Bensmaia, S.J. (2015a). Sensitivity to microstimulation of somatosensory cortex distributed over multiple electrodes. Frontiers Syst Neurosci 9, 47.

Kim, S., Callier, T., Tabot, G.A., Gaunt, R.A., Tenore, F.V., and Bensmaia, S.J. (2015b). Behavioral assessment of sensitivity to intracortical microstimulation of primate somatosensory cortex. Proc National Acad Sci 112, 201509265.

Marks, L.E., Szczesiul, R., and Ohlott, P. (1986). On the Cross-Modal Perception of Intensity. J Exp Psychology Hum Percept Perform 12, 517–534.

Michelson, N.J., Eles, J.R., Vazquez, A.L., Ludwig, K.A., and Kozai, T.D.Y. (2019). Calcium activation of cortical neurons by continuous electrical stimulation: Frequency dependence, temporal fidelity, and activation density. J Neurosci Res 97, 620–638.

Overstreet, C.K., Klein, J.D., and Tillery, S.I.H. (2013). Computational modeling of direct neuronal recruitment during intracortical microstimulation in somatosensory cortex. J Neural Eng 10, 066016.

Pei, Y.-C., Denchev, P.V., Hsiao, S.S., Craig, J.C., and Bensmaia, S.J. (2009). Convergence of submodality-specific input onto neurons in primary somatosensory cortex. J Neurophysiol 102, 1843 1853.

Prsa, M., Morandell, K., Cuenu, G., and Huber, D. (2019). Feature-selective encoding of substrate vibrations in the forelimb somatosensory cortex. Nature 567, 384–388.

Stoney, S.D., Thompson, W.D., and Asanuma, H. (1968). Excitation of pyramidal tract cells by intracortical microstimulation: effective extent of stimulating current. Journal of Neurophysiology 31, 659 669.

Sur, M., Wall, J.T., and Kaas, J.H. (1984). Modular distribution of neurons with slowly adapting and rapidly adapting responses in area 3b of somatosensory cortex in monkeys. Journal of Neurophysiology 51, 724 744.